



# Insights into tropical cloud chemistry at Reunion Island (Indian Ocean): results from the BIO-MAÏDO campaign

Pamela A. Dominutti[1*], Pascal Renard[1], Mickaël Vaïtilingom[2], Angelica Bianco[1], Jean-Luc Baray[1,3], Agnès Borbon[1], Thierry Bourianne[4], Frédéric Burnet[4], Aurélie Colomb[1], Anne-Marie Delort[5], Valentin Duflot[6], Stephan Houdier[7], Jean-Luc Jaffrezo[7], Muriel Joly[5], Martin Leremboure[5], Jean-Marc Metzger[8], Jean-Marc Pichon[3], Mickaël Ribeiro[1], Manon Rocco[1], Pierre Tulet[6,9], Anthony Vella[7], Maud Leriche[1,10] and Laurent Deguillaume[1,3*]

[1] Laboratoire de Météorologie Physique, UMR 6016, CNRS, Université Clermont Auvergne, 63178 Aubière, France

[2] Laboratoire de Recherche en Géosciences et Energies, EA 4539, Université des Antilles, 97110 Pointe-à-Pitre, France

[3] Observatoire de Physique du Globe de Clermont-Ferrand, UAR 833, CNRS, Université Clermont Auvergne, 63178 Aubière, France

[4] Centre National de Recherches Météorologiques (CNRM), Université de Toulouse, Météo-France, CNRS, Toulouse, France

[5] Institut de Chimie de Clermont-Ferrand, CNRS, SIGMA Clermont, Université Clermont Auvergne, 63178 Aubière, France

[6] Laboratoire de l'Atmosphère et des Cyclones (LACy), UMR 8105, Université de la Réunion-CNRS-Météo-France, Saint-Denis de La Réunion, France

[7] Université Grenoble Alpes, CNRS, IRD, IGE (UMR 5001), 38000 Grenoble, France

[8] Observatoire des Sciences de l'Univers de La Réunion (OSUR), UAR 3365, Saint-Denis de la Réunion, France

[9] Laboratoire d'Aérologie, UMR 5560 (Université de Toulouse, CNRS, IRD), Toulouse, France

[10] Centre pour l'étude et la simulation du climat à l'échelle régionale (ESCER), Département des sciences de la terre et de l'atmosphère, Université du Québec à Montréal, Montréal, Canada

*Correspondence to:* Laurent Deguillaume (laurent.deguillaume@uca.fr) and Pamela Dominutti (pamela.dominutti@uca.fr)

**Abstract.**

We present here the results obtained during an intensive field campaign conducted in the framework of the French "BIO-MAÏDO" (Bio-physico-chemistry of tropical clouds at Maïdo (Réunion Island): processes and impacts on secondary organic aerosols' formation) project. This study integrates an exhaustive chemical and microphysical characterization of cloud water obtained in March-April 2019 at Reunion Island (Indian Ocean). Fourteen cloud samples have been collected along the slope of this mountainous island. A comprehensive chemical characterization of these samples is performed, including inorganic ions, metals, oxidants, and organic matter (organic acids, sugars, amino acids, carbonyls, and low-soluble volatile organic compounds (VOCs)). Cloud water presents high molecular complexity with elevated water-soluble organic matter content partly modulated by microphysical cloud properties.

As expected, our findings show the presence of compounds of marine origin in cloud water samples (*e.g.*, chloride, sodium) demonstrating ocean–cloud exchanges. However, the non-sea salt fraction of sulphate varies between 38 and 91%, indicating the presence of other sources. Also, the presence of amino acids and for the first time in cloud waters of sugars, clearly indicates that biological activities contribute to the cloud water chemical composition.

A significant variability between events is observed in the dissolved organic content ($25.5 \pm 18.4$ mgC L$^{-1}$), with levels reaching up to 62 mgC L$^{-1}$. This variability was not similar for all the measured compounds, suggesting the presence of dissimilar emission sources or production mechanisms. For that, a statistical analysis is performed based on back-trajectory calculations using the CAT (Computing Atmospheric Trajectory Tool) model associated with land cover registry. These investigations reveal that air mass





origins and microphysical variables do not fully explain the variability observed in cloud chemical composition, highlighting the complexity of emission sources, multiphasic transfer, and chemical processing in clouds.

Additionally, several VOCs (oxygenated and low-soluble VOCs) were analysed in both gas and aqueous phases. Significant levels of biogenic low-soluble VOCs were detected in the aqueous phase, indicating the cloud-terrestrial vegetation exchange. Cloud scavenging of VOCs is assessed and compared to Henry's law equilibrium to evaluate potential super or sub saturation conditions. The evaluation reveals the supersaturation of low-soluble VOCs from both natural and anthropogenic sources. Our results depict even higher supersaturation of terpenoids, suggesting their importance in the aqueous phase chemistry in highly impacted tropical

areas.

**Keywords: Cloud chemical composition, dissolved organic carbon, gas/aqueous phase partitioning, tropical atmosphere, air mass influence.**

**1.    Introduction**

The chemical composition of the atmosphere modulates its impacts on the global climate, regional air pollution, human health, and ecosystems (Monks et al., 2009; Seinfeld and Pandis, 2006). Inorganic and organic compounds are abundant within the three atmospheric compartments: gases, aerosol particles and clouds (liquid water and ice); they are emitted from a variety of natural and anthropogenic sources. When transported in the atmosphere, away from source regions, these compounds undergo numerous

multiphasic chemical transformations. These include (i) homogeneous photochemical reactions in the gaseous phase (ii) gas to particle conversion (nucleation, condensation on pre-existing particles) (iii) dissolution processes in the aqueous phase during cloud or fog events and subsequent aqueous phase reactivity, possibly followed by return to the atmosphere, and (iv) removal by wet deposition. Clouds are known to play an important role in atmospheric chemistry, affecting the homogeneous and heterogeneous gas-phase chemistry (McNeill, 2015). Indeed, cloud droplets can dissolve soluble gases and soluble part of aerosol particles acting

as cloud condensation nuclei (CCN). They represent aqueous chemical reactors where chemical and biological transformations occur, acting as source or sink of chemical species and altering their distribution among the various atmospheric phases (Herrmann et al., 2015). Clouds will consequently modify the physico-chemical aerosol properties (oxidation state, chemical composition, hygroscopicity).

Since the late 1990s, substantial efforts have been made to further understand the chemical composition and reactivity of dissolved

matter in cloud droplets through in situ measurements and laboratory investigations (Aleksic et al., 2009; Bianco et al., 2017; Brege et al., 2018; Brüggemann et al., 2005; Cini et al., 2002; Deguillaume et al., 2014; Fomba et al., 2015; Gilardoni et al., 2014; Gioda et al., 2013; Herckes et al., 2013, 2002; Lee et al., 2012; Li et al., 2020; van Pinxteren et al., 2016, 2020; Renard et al., 2020; Vaïtilingom et al., 2013; Viana et al., 2014). Some of these studies were focused on the role of clouds in the lifecycle of inorganic compounds (*i.e.,* ions, metals, and oxidants). The ionic composition of clouds is of substantial importance since, for example, (i) it

controls the cloud water acidity (Pye et al., 2020), ii) clouds represent the most important formation pathways for sulphate (Chin et al., 2000), and (iii) it helps to assign air masses origin (Renard et al., 2020). The characterization of strong oxidants, such as hydrogen peroxide and transition metal ions, is also crucial since they participate to the oxidative capacity of the cloud water through production of HO• radicals (Bianco et al., 2015).

Quantifying the organic matter mass distribution and composition in the different atmospheric compartments will improve the

understanding and characterization of their impacts on air quality and climate. The organic carbon fraction of all atmospheric compartments comprises a large range of compounds and structures (Cook et al., 2017; Zhao et al., 2013). The dissolved organic carbon (DOC) not only contributes to the hygroscopic properties of aerosols but also to aqueous cloud chemistry (Herckes et al., 2013). In recent years, an increasing number of studies approached the characterization of organic matter and DOC and its processing by fogs and clouds. It has been shown that fogs and clouds contain a significant amount of dissolved organic matter, ranging from

1 to as much as 200 mgC L$^{-1}$ depending on the environmental conditions (Herckes et al., 2013). The highest values were observed



near urban conglomerates and in clouds impacted by biomass burning while the lowest levels were observed in marine/remote environments (Herckes et al., 2013). However, most of these studies were using targeted analytical approach which focused on small chain carboxylic acids, dicarboxylic acids, carbonyls (Deguillaume et al., 2014; Löflund et al., 2002). These compounds have been selected for several distinct reasons, non-exhaustive including: they are present in significant concentrations, they are

representative of different sources (Rose et al., 2018), they are likely to contribute to secondary organic aerosols in cloud and humid aerosol (aqSOA) production by aqueous phase reactivity (Ervens et al., 2011). However, a very large fraction of DOC in clouds remain uncharacterized, despite the application of a variety of targeted analytical approaches (Bianco et al., 2016; Herckes et al., 2013). Recently non targeted approaches using high resolution mass spectrometry have been deployed to fully characterize the organic matter present in cloud water (Bianco et al., 2018; Cook et al., 2017; Zhao et al., 2013). These studies have revealed the

complexity of the matrices with thousands of different compounds resulting from biogenic or anthropogenic sources, and secondary products from atmospheric reactivity (Bianco et al., 2019). However, these global characterizations are still not quantitative.

Given the complexity and the evolving nature of clouds, their measurement and chemical characterization represent a challenge and have resulted in significant uncertainties in our understanding of formation/evolution processes, chemical composition, fate, as well as potential impacts. Some well-established measurement observatories located at high altitudes have been carried out in numerous

studies about cloud chemical composition, such as puy de Dôme (PUY) observatory in France (Bianco et al., 2017; Deguillaume et al., 2014; Renard et al., 2020; Vaïtilingom et al., 2013; Wang et al., 2020), Schmücke Mountain in Germany (Brüggemann et al., 2005; Herrmann et al., 2005; van Pinxteren et al., 2005, 2016; Roth et al., 2016; Whalley et al., 2015), East Peak in Puerto Rico (Gioda et al., 2013), Mount Tai in China (Li et al., 2017; Liu et al., 2012a; Shen et al., 2012), Whiteface Mountain New York (Aleksic et al., 2009; Dukett et al., 2011; Lance et al., 2020), Po Valley (Brege et al., 2018; Gilardoni et al., 2014) and Tai Mo Shan

in Hong Kong SAR (Li et al., 2020). Nevertheless, most of the aforementioned observatories are located in the mid-northern latitudes with significant influence of continental emissions from natural and anthropogenic origins. A lack of measurements is observed in the southern hemisphere, at tropical latitudes and in remote marine environments. This raises some questions on the chemical composition of clouds in these regions presenting highly different environmental conditions (temperature, sun irradiation, emission sources, *etc.*) and on the impacts that these conditions can have on the distribution and transformations of the DOC in cloud water.

Reunion Island situation in the tropics offers a unique context to analyse the kinetics and photochemical processing under cloudy conditions. Moreover, due to its geographic location, climate and topography conditions, the island provides an excellent framework to study multiphasic chemical formation processes.

We present here the results obtained during an intense field campaign conducted in the framework of the French BIO-MAÏDO project. Our study evaluates the chemical composition and physical properties of fourteen cloud events collected at Reunion Island

in March-April 2019. A highly comprehensive chemical screening has been conducted in cloud water samples including ions, metals, oxidants, and organic matter (organic acids, sugars, amino acids, carbonyls, and low-soluble volatile organic compounds (VOCs)). We address here the questions concerning the variability of the chemical composition and their sources from the data obtained in several episodes, in relation to the physicochemical and meteorological characteristics. Finally, cloud scavenging of volatile organic compounds is assessed and compared to Henry's law equilibrium to evaluate possible super or subsaturation

conditions.

## 2. Materials and Methods

The BIO-MAÏDO field campaign was conducted between 14 March and 4 April 2019 at Reunion Island. During this period, several sites along the mountain slope of the Maïdo region were instrumented to evaluate chemically, physically, and biologically the different atmospheric phases and their processing (aerosol, gas, and clouds).

Reunion is a small volcanic tropical island located in the south-western Indian Ocean, affected by south- easterly trade winds near the ground and westerlies in the free troposphere (Baray et al., 2013). The island encompasses 100 000 ha of native ecosystems and it is located far from the impact of large anthropogenic emission sources (Duflot et al., 2019). Due to its abrupt topography, the





collection of cloud water samples developing on the slope of the mountains can be conducted in a regular basis. The location of the island provides an original and unexplored setting to assess the combination of biogenic and marine sources and the potential

interactions between both emissions

Reunion island is characterised by a complex atmospheric dynamic. During the night and early morning, air masses on sites at high altitude are separated from local and regional sources of pollution, due to the strengthening of the large-scale subtropical subsidence at night (Lesouëf et al., 2011, 2013). The cloud sampling point is located in the dry west region of the island and mainly surrounded by biogenic sources as tropical forests characterised by endemic tree species *Acacia heterophylla (Fabaceae)* and, plantations of

the coniferous species *Cryptomeria japonica (Taxodiacae)*; the *Acacia heterophylla* forest which is locally called "Tamarinaie" (Duflot et al., 2019). The place was strategically selected as clouds develop daily on the slopes of the Maïdo region, with a well-established diurnal cycle (formation in the late morning, dissipation at the beginning of the night (Baray et al., 2013; Lesouëf et al., 2011). Generally, these slope clouds over the Maïdo region are characterized by low vertical development and low water content. This characteristic is also conducive to the aqueous transformation of organic compounds: more UV radiation in the core of the

cloud and low wet deposition by rainfall.

### 2.1 Sampling strategy

The sampling strategy was designed to study the chemical exchanges between the different atmospheric compartments and to evaluate the influence of sources and atmospheric dynamics on the atmosphere's composition.

Cloud sampling was performed at "Piste Omega" (21°03'26"S, 55°22'05"E) located at 1760 m above sea layer (a.s.l.) (Figure 1). The sampling site is located on the western part of the island, along the road to Maïdo peaks, around 5 kilometres away from the Maïdo observatory (Baray et al., 2013). Cloud samples were obtained using a cloud collector (Deguillaume et al., 2014; Renard et al., 2020) which collects cloud droplets larger than 7 µm (estimated cut-off diameter) by impaction onto a rectangular aluminium plate. The cloud collector was installed at the top of a 10m mast and cloud water samples were obtained on event basis. Before

sampling, the cloud collector was rinsed thoroughly with Ultrapure-MiliQ water and the aluminium plates, the funnel and the container were autoclaved to avoid any chemical and biological contamination. Procedural controls were obtained by spreading MilliQ water on the clean collector. The control samples were analysed to document chemical and biological contamination. Once cloud samples were obtained, pH was determined, and the samples were filtered using a 0.20 µm nylon filter to eliminate microorganisms and micrometric particles. Cloud samples were aliquoted in vessels (plastic polypropylene, borosilicated glass) the

adapted the chemical target, then refrigerated or frozen and stored until subsequent chemical analyses.

In total, fourteen cloud episodes (noted R1 to R14) during thirteen different days were collected between 14 March and 4 April 2019, with variable volumes ranging from 8 to 138 mL, and an average volume of 59 ± 39 mL. For most of the samples, the volume was insufficient to all the planned targeted chemical analyses, especially for the organic characterization. Table S3 reports the physical and chemical characteristics of the cloud events, such as: dates, sampling period, pH, liquid water content (LWC), mean

effective diameter (Deff), temperature and concentrations of chemical species: ions, oxidants ($H_2O_2$, Fe(II), Fe(III)), trace elements, sugars, mono and dicarboxylic acids, amino acids, oxygenated VOCs (carbonyls), VOCs, TC, IC and TOC concentrations.

### 2.2 Chemical analyses

The complete chemical screening was determined by the volume of the samples but included for most of the samples the

identification and quantification of trace metals, ions, oxidants, and dissolved organic matter such as organic carbon (DOC), sugars, carbonyls, organic acids, low-soluble VOCs and amino acids. Gaseous organic compounds were also measured during the cloud events with the aim to quantify the VOCs partitioning between the gas and aqueous phases. Table S1 presents an overview of the analysis performed on each cloud water sample. A condensed summary of the analytical procedures used for the analysis are recalled below and more details about the protocols and uncertainties evaluation are given in the Supplementary Information (SI1). Limits

of detection, limits of quantification and uncertainties for each species are described in Table S2.





### Metals and oxidants in cloud water

Abundances of trace metals are analysed using an Inductively Coupled Plasma Mass Spectrometry instrument (ICP-MS, Agilent 7500). The cloud water collector being made of aluminium, this element is not quantified in the samples. More details about the
instrument conditions and the analytical technique used can be found in Bianco et al. (2017).

Hydrogen peroxide and iron concentrations are analysed by UV-visible spectroscopy, applying derivatization techniques. Hydrogen peroxide in cloud water is quantified with a miniaturized Lazrus fluorimetric assay (Vaïtilingom et al., 2013; Wirgot et al., 2017). Fe(II) concentration is quantified by UV-visible spectroscopy ($\lambda$=562 nm), based on the rapid complexation of iron with ferrozine (Stookey, 1970). The total iron content (Fe(tot)) is detected after the reduction to Fe(II) by the addition of aspartic acid. Fe(III)
concentration is then determined by deducing the concentration of Fe(II) from Fe(tot) (Parazols et al., 2006).

Total carbon (TC) and inorganic carbon (IC) are measured with a Shimadzu TOC-L analyser. Total organic carbon (TOC) quantification is then obtained by the difference between the measured TC and IC.

### Major inorganic and organic ions in cloud water

Diluted cloud samples are analysed by ion chromatography allowing the quantification of the major organic and inorganic ions
(acetic, formic, and oxalic acids, $Cl^-$, $NO_3^-$, $SO_4^{-2}$, $Na^+$, $K^+$, $NH_4^+$, $Mg^{2+}$ and $Ca^{2+}$, MSA and $Br^-$). More details about the analytical method for ion chromatography analysis has been previously reported by Jaffrezo et al. (1998) and Bianco et al. (2018).

### Organic matter in cloud water

Anhydro sugars, sugar alcohols, and primary saccharides are analysed by HPLC with amperometric detection (PAD), allowing to quantify anhydrous saccharides (levoglucosan, mannosan, galactosan), polyols (arabitol, sorbitol, mannitol, erythritol, and xylitol),
glucose and trehalose (Samaké et al., 2019b; Waked et al., 2014).

The analysis of a large array of organic acids (including pinic, phthalic acids, and 3- MBTCA (3-methyl-1,2,3-butanetricarboxylic acid)) is conducted using the same water extracts as for IC and HPLC-PAD analyses. This is performed by HPLC-MS with negative mode electrospray ionization (Borlaza et al., 2021).

Carbonyl compounds are analysed after derivatization by fluorescent dansylacetamidooxyamine (DNSAOA), an oxyamino reagent
which is specific for carbonyl compounds (Houdier et al., 2000). Derivatization reactions are performed in the presence of anilinium chloride (AnCl) as a catalyst (Houdier et al., 2018). This original approach, coupling both AnCl catalysed derivatization and the use of HPLC-MS allows us to quantify (i) single aldehydes (formaldehyde (F) and acetaldehyde (A)), (ii) polyfunctional aldehydes (hydroxyacetaldehyde (HyA), glyoxal (GL), and methylglyoxal (MGL)) and (iii) ketones (acetone (AC) and hydroxyacetone (HyAC)). To the best of our knowledge, this work provides the first measurements of HyAc in environmental water samples.

A total of fifteen amino acids (Ala, Arg, Asn, Asp, Gln, Glu, Gly, His, Lys, Met, Phe, Ser, Thr, Trp, Tyr) are quantified in twelve cloud samples (R1, R2, R3, R4, R5, R7, R8, R9, R10A, R10B, R11, R13) by UPLC-HRMS (Ultra High-Performance Liquid Chromatography coupled with High Resolution Mass Spectrometry). As the standard addition method is used for the quantification, twelve samples ready for UPLC-HRMS analysis are prepared containing the original cloud water added with 19 AAs at final concentrations set to 0.1, 0.5, 1.0, 5.0, 10, 25, 50, 100, 150, 500 µg $L^{-1}$. Chromatographic separation of the analytes is performed on
BEH Amide/HILIC column, and the MS analysis with a Q Exactive™ Hybrid Quadrupole-Orbitrap ™ Mass Spectrometer, the Q-Exactive ion source is equipped with electrospray ionization (ESI+). More details about the analytical method, calibration curves and quality control procedures can be found in Renard et al. (2021).

Hydrophobic VOCs are extracted by Stir Bar Sorptive Extraction (SBSE) and analysed by a thermal desorber gas chromatograph coupled to a mass spectrometer (SBSE-TD-GCMS), following the optimized procedure described in Wang et al. (2020). SBSE is
used to extract the VOC from the aqueous phase thanks to 126 µL-stir bars coated with polydimethylsiloxane (PDMS). Extraction efficiencies by SBSE vary between 22 and 97%. A total of twelve aromatics and terpenoids have been detected in Reunion cloud samples.



*Gaseous organic matter*

During the cloud collection, gaseous Oxygenated VOCs (OVOC) and VOCs were simultaneously sampled for a total of seven cloud events (Table S1) with the new AEROVOCC sampler (AtmosphERic Oxygenated/ Volatile Organic Compounds in Cloud). AEROVOCC collects gaseous OVOCs/VOCs by deploying simultaneously two types of sorbent tubes: Tenax® TA sorbent tube and PFBHA (pentafluorobenzylhydroxylamine) pre-coated Tenax® TA sorbent tubes. Tenax® is chemical inert, highly hydrophobic, and widely used for VOCs measurements of more than four carbon atoms (Ras et al., 2009; Schieweck et al., 2018). PFBHA and MTBSTFA are derivatization agents specific for –C=O and –OH/-COOH functional groups, respectively. Cloud

sampling is performed simultaneously with the cloud collector presented above. Tenax® TA sorbent tubes are installed at the same altitude as the cloud collector with the inlet facing downward. The whole set of tubes is placed into a stainless-steel funnel. Each tube is connected to a Gilair *Plus* pump (Gilian) at ground level. The three pumps are placed into a waterproof Pelicase case, and provide a controlled flow rate of 100 mL min$^{-1}$ for a sampling duration of 40 min. Before sampling, sorbent tubes were pre-conditioned for 5 h at 320°C under a nitrogen flow of 100 mL min$^{-1}$. These conditions guarantee the presence of the target

OVOCs/VOCs in the blank at levels lower than the detectable mass by the TD-GC-MS, which was used for the analysis of all samples. More details on analytical conditions can be found in Dominutti et al. (2019) and Wang et al. (2020) for low-soluble VOC. The preparation of the tubes and the analytical conditions for OVOCs are described in the SI and are adapted from Rossignol et al. (2012).

**2.3 Physical parameters**

Microphysical parameters were measured at the sampling point during the cloud sampling. Droplet size distribution measurements were performed by a cloud droplet probe (CDP, Droplet Measurement Technologies (DMT)) which was attached to the mast just below the cloud water collector (Figure 1).

The CDP, described in Lance et al. (2010), provides 1 Hz cloud droplet size distribution on 30 bins from 2 to 50 μm in diameter.

Originally designed to equipped research aircraft, this probe has been adapted for use in a fix point. A small fan fixed just to the rear of the laser beam creates an air flow of 5m s$^{-1}$ in the sampling section. This value has been empirically determined from comparison with reference instruments such as the Fog-Monitor also manufactured by DMT. This system avoids the use of an inlet which usually introduces instrumental bias when the wind direction deviates from the inlet axis (Guyot et al., 2015; Spiegel et al., 2012). The liquid water content (LWC), that is the mass of water droplet per unit of volume of air, is derived from the size

distribution: $LWC = \frac{\pi}{6} * \rho_w * \sum(Ni * Di^3)$, where $Ni$ is the droplet number concentration in the bin size $i$, $Di$ is the droplet diameter at the centre of the bin $i$, and $\rho_w$ is the water density. The effective diameter is calculated as: $Deff = \sum(Ni * Di^3) / \sum(Ni * Di^2)$. Ancillary data also include the measurements of meteorological variables at the sampling point. Air temperature, relative humidity and wind speed were measured every 10 seconds during the field campaign with a PT100, a Vaisala HMP110 and a Young 12102 3-cup anemometer, respectively.


**2.4 Dynamical analyses**

Air masses backward trajectories are calculated with the CAT (Computing Atmospheric Trajectory Tool) model (Baray et al., 2020). A high-resolution version of CAT (MesoCAT) has been developed especially for the BIO-MAÏDO project (Rocco et al., 2021). MesoCAT assimilates wind and topography fields from the mesoscale non hydrostatic model Meso-NH (Lac et al., 2018). For this

work, we use Meso-NH 3D outputs every one-hour from a domain using a horizontal resolution of 100 meters. Details of Meso-NH simulation are given in Rocco et al. (2021). In this work, backward trajectories from Piste Omega (the cloud sampling site) are computed with MesoCAT for each day when cloud samples were collected. Trajectories are calculated every 15 minutes from the beginning to the end of the sampling. The MesoCAT configuration includes 75 points of trajectories in a starting three-dimensional domain of 100m × 100m × 50m (latitude, longitude, altitude) around Piste Omega, and the temporal resolution and duration of back





trajectories are 5 min and 12 hours, respectively. The atmospheric water vapour and the cloud content parameters provided by Meso-NH are interpolated on each trajectory point.

Additionally, in order to estimate the influence of the soil type located under the trajectory points, an interpolation of the land zones was done using the Corinne Land Cover 2018 (UE – SOeS, CORINE Land Cover, 2018, Geoportail, https://www.geoportail.gouv.fr/). In total, fifty categories of land use are detailed on the land registry. To evaluate the back

trajectories during the cloud sampling, a merge of the land registry into 4 categories is proposed here: vegetation, urban area, coastal area, and farming area. Sensibility tests between 300 and 1000m above ground/sea level are performed to select the maximum altitude of back trajectories. Only back trajectory points lower than 500 m above sea or ground level are considered to be influenced by the surface and are then taken into account in the statistics to evaluate the potential contribution of emission sources on cloud chemical composition.


### 3. Results

#### 3.1 Physical parameters

Measurements of LWC and Deff were carried out during the cloud water sampling. To evaluate these microphysical parameters together with the chemical content of clouds, a "pre-treatment" was performed. To this end, CDP data were first filtered excluding

the breaks effectuated during cloud sampling and considering a cut-off diameter of 7 µm of the cloud collector. Figure2 shows the variability of LWC and Deff of droplets observed during each cloud event of this campaign. LWC exhibits a limited variation, with an average value of $0.07 \pm 0.04$ g m$^{-3}$, ranging between 0.02 and 1.07 g m$^{-3}$. The average values present lower LWC levels than those observed in previous campaigns under marine influence: $0.19 \pm 0.16$ g m$^{-3}$ (0.05-0.92 g m$^{-3}$ for highly marine clouds) at PUY, $0.17 \pm 0.13$ (0.02-0.68) g m$^{-3}$ at Puerto Rico and $0.29 \pm 0.10$ (0.11-0.46) g m$^{-3}$ at Cape Verde (Gioda et al., 2013; Renard et al.,

2020; Triesch et al., 2021). The lower LWC values observed could be related to the local atmospheric and geographical conditions, which affect the cloud formation processes over the island. The development of diurnal thermally induced circulations, combining downslope (catabatic winds) and land breezes at night, and upslope (anabatic winds) and sea breeze during the day-time, causes a quasi-daily formation of clouds, which are usually weakly developed vertically with a low water content (Duflot et al., 2019). Some small differences between cloud events can be noted, with particularly low LWC values observed during the R3, R4 and R12

episodes. The relationship between LWC and inorganic concentrations in cloud water has been previously addressed showing the influence that an increase of LWC could have on the dilution of solutes in cloud water (Aleksic and Dukett, 2010). Nevertheless, this influence is not significantly observed in our study since no clear relationship was found between LWC and chemical concentrations such as total ion content TIC and total organic carbon TOC. It is important to have in mind the limited number of cloud samples collected in our study, which could not provide enough basis for concluding about LWC effects on cloud chemistry.

The effective diameter of cloud droplets was also measured. Average diameters of $13.7 \pm 1.51$ µm were observed, with slight variations within the cloud events. Despite the small variations in LWC and Deff between events, the time series of the microphysical parameters and cloud occurrence were strongly different among the episodes (Figure S2).

Correlations between TOC *vs* LWC and Deff were investigated in order to explore the relationship between chemical content and microphysical conditions. No correlation between TOC and LWC/Deff is observed from our measurements (Figure S3). This result

suggests that microphysical parameters are not the main parameter regulating the organic content in cloud water, and other processes and sources are responsible for the observed levels of dissolved compounds. Further analysis about the air masses trajectories, physical variables and chemical tracers will be discussed in the section 4.

#### 3.2 Inorganic chemical composition

***Inorganic ions***





Figure 3 reports the main inorganic ion concentrations (µM) and the relative contribution of all species for the fourteen cloud events collected at Reunion Island. On average, the most abundant anions are $Cl^-$ (434 ± 370 µM) and $NO_3^-$ (239 ± 168 µM), and the most abundant cations are $Na^+$ (490 ± 399 µM) and $NH_4^+$ (123 ± 43 µM). Naturally, the ionic composition of clouds in Reunion Island reflects the major marine influence due to the high concentrations of sea salt ($Na^+$, $Cl^-$) and the presence of other ions (such as $SO_4^{2-}$

) that could also originate in the atmosphere from the marine surface. $Na^+$ and $Cl^-$ dominate the inorganic composition contributing to 30% and 27% respectively, to the average total ion content. Nitrate is the third largest contributor (15%), followed by ammonium (7.6%) and sulphate (7.3%). Chloride and sodium concentrations were similar to those reported in previous studies performed at marine sites such as in Puerto Rico (384-473 µM for $Cl^-$ and 362-532 µM for $Na^+$) (Gioda et al., 2009, 2011; Reyes-Rodríguez et al., 2009), but lower sodium levels than those recently observed at Cape Vert (870 ± 470 µM) (Triesch et al., 2021). Some mid-

latitude remote sites such as the PUY observatory are regularly under marine influence and can present elevated $Cl^-$ and $Na^+$ concentrations especially for air masses classified as "highly marine" in the study from Renard et al. (2020). However, the $Cl^-$ and $Na^+$ concentrations in our study are on average respectively 2.6 and 2.5 times higher than those observed at PUY (for highly marine clouds). The difference could be mainly explained by the remoteness of PUY to marine environment and/or the dilution effect at PUY (higher LWC and mean diameter, which modulates the concentrations of ions).

$NO_3^-$ concentrations present high average concentrations at Reunion Island than those reported in other cloud field campaigns under marine influence. Nitrate levels are 8 to 20 times higher than those observed in cloud water at marine sites and 4 times higher than at PUY for "highly marine" clouds (Gioda et al., 2009; Renard et al., 2020; Reyes-Rodríguez et al., 2009). The detection of elevated nitrate concentrations in cloud water implies the influence of local anthropogenic sources; its contribution could be associated with the uptake of gaseous $NO_x$/nitric acid, and/or from the dissolution of nitrate from aerosols (Benedict et al., 2012; Leriche et al.,

2007). The observed concentrations of $NH_4^+$ are lower than those observed in cloud water from polluted areas (Guo et al., 2012) but higher or similar than those measured for remote sites in Central America and Europe (Deguillaume et al., 2014; Gioda et al., 2013; van Pinxteren et al., 2016). This reflects the possible influence of terrestrial/agricultural sources (gaseous ammonia and ammonium present on aerosol). For $SO_4^{2-}$, average concentrations observed in this study (118.5 ± 44.04 µM) are higher than those reported for highly marine clouds at PUY-France and at East Peak-Puerto Rico, by factors of 3.5 and 4.4, respectively (Gioda et al., 2011; Renard

et al., 2020). Nevertheless, similar sulphate average concentrations were observed at a marine site in Cape Verde (116 µM) and at Tai Mo Shan in Hong Kong (152 µM) (Li et al., 2020; Triesch et al., 2021). It can be noted that non-sea salt sulphate estimated in cloud events represent 76.2 ± 14.1% of the total sulphate measured during our study. Thus, differences observed on sulphate levels with other marine sites could indicate the contribution of additional anthropogenic sources.

The correlation and the concentration ratios between ionic species are useful to understand the air masses origin and the atmospheric

processes involved. The predominant species, $Na^+$ and $Cl^-$, present a strong correlation ($r^2 = 0.87$), suggesting similar air masses origin (Figure S4). However, the average $Cl^-/Na^+$ ratio (0.85) is lower than the sea-salt molar ratio (1.17, Holland, (1978), Figure 3b). Ratios lower than 1 reflect chloride depletion, possibly associated with the sea-salt reaction with strong acids such as $HNO_3$ and $H_2SO_4$. This chloride depletion from sea salt particles had already been observed and was associated with the presence of $NO_x$ and $SO_2$ sources (Benedict et al., 2012; Gioda et al., 2009; Li et al., 2020). Regarding $SO_4^{2-}/Na^+$, exceedances of the standard sea-salt

molar ratio are observed (0.06 (Holland, 1978), by a factor of 2 to 8, Figure 3b) confirming the influence of non-sea salt sources. It is well-known that the greater part of sulphate in the atmosphere (70%) is formed in cloud droplets, and only a small portion is produced from condensation of $H_2SO_4$ on particles or from formation in aqueous aerosol (Ervens et al., 2011; Textor et al., 2006). Thus, the sources of cloud water sulphate may include uptake of gaseous $SO_2$ followed by its later aqueous oxidation to $H_2SO_4$ in cloud droplets. $SO_2$ sources include direct emissions from anthropogenic sources (shipping, vehicles, power plants emissions) and

natural emissions (volcanoes and DMS oxidation) (Gondwe et al., 2003). The sulphate in clouds could also be associated with the scavenging of aerosol sulphate (frequently in the form of ammonium sulphate). Nevertheless, the ratio between $nss-SO_4^{2-}$ and $NH_4^+$ presents average values of 1.73 ± 1.28 in molar basis. This result corroborates the excess of sulphate in the aqueous phase, suggesting the contribution of other sources than aerosols on the clouds observed.


Other ions present in our samples ($Mg^{2+}$ and $K^+$) correlate well with $Na^+$ ($r^2 = 0.87, 0.65$, respectively). Magnesium to sodium ratios present lower values (0.07) than those expected from seawater (0.23), suggesting the depletion of $Mg^{2+}$ (Figure S4). Contrarily, the $K^+$ to $Na^+$ show an enrichment of expected ratios (0.08 *vs* 0.02); a possible explanation could be the influence of biomass burning on cloud water (Urban Cerasi et al., 2012) (Figure S4). Bromide and MSA present elevated correlations with sodium (0.86 and 0.73, respectively) indicating the relation of these ions with seawater source. However, their concentrations are quite low resulting in a small contribution to average ion total mass. Similar results have been obtained in other studies near sea sources, where MSA did

not display an important contribution between the ionic species (Benedict et al., 2012; van Pinxteren et al., 2020).

An enrichment of calcium relative to seawater is also observed (0.06 *vs* 0.04) with lower correlation coefficients ($r^2 = 0.64$). The excess of $Ca^{2+}$ was already observed in cloud water (Benedict et al., 2012; Straub et al., 2007), which may be associated with the soil contribution.

*Trace metals and hydrogen peroxide*

The trace metal concentrations observed are very low, with some species frequently below the limit of quantification. It is important to note that cloud samples were filtered using a 0.2 µm filter, removing the insoluble fraction of aerosols. Figure S5 reports the trace metal concentrations for each cloud sample separated by concentration range. Those compounds can originate from crustal dust or particles from marine or anthropogenic sources. Mg and Zn are the most concentrated elements, observed in all the samples and

ranged from 14.1 to 3.32 and 1.85 to 0.07 µM, respectively. Even though their low concentrations Cu, Mn, Ni, Sr, Fe and V are between the most abundant trace metals observed. Mg and Zn come mainly from natural origin such as soil dust or sea salt. Zn can particularly be enriched in sea salt during bubble bursting (Piotrowicz et al., 1979) and has been found to be dominant in marine environments (Fomba et al., 2013). Some studies investigated the level of trace metals for marine environments and reported low concentration values for clouds (Vong et al., 1997) and aerosol particles (Fomba et al., 2013, 2020). Similar levels and distribution

of trace metals have been also reported at the PUY that is importantly influenced by marine emissions (Bianco et al., 2015). The low levels observed in this study also demonstrate the low inputs of trace metals from anthropogenic sources.

$H_2O_2$ levels are examined on 12 clouds of this study (Figure S6). This oxidant dissolves from the gas phase to the aqueous phase (main source) (McElroy, 1986; Xuan et al., 2020) but it can also be produced through aqueous phase chemistry (Mouchel-Vallon et al., 2017). Concentrations of gaseous $H_2O_2$ have been reported in few studies in the last years (Fischer et al., 2015; Junkermann and

Stockwell, 1999; O'Sullivan et al., 1999; Weller et al., 2000). These studies have shown a latitudinal variation of $H_2O_2$ concentrations with higher values observed in the tropics (>500 ppt) and decreasing when the latitude increase in both hemispheres (reaching ~ 250 ppt at 40º of southern hemisphere) (Fischer et al., 2015). The $H_2O_2$ aqueous phase photo-reactivity is particularly suspected to produce hydroxyl radicals through its photolysis and its reactivity with iron (Fenton processes) (Bianco et al., 2015, 2020), contributing by this way to the oxidative capacity of cloud droplets. Measured values indicated in this study could be biased due to

the storage of samples and so should be carefully analysed. Values range from 0.04 to 8.79 µM with an average value of $1.94 \pm 2.57$ µM and are surely underestimated. However, those values are interesting since they highlight high variability among the samples indicating different photochemical equilibrium. Globally, we can notice that $H_2O_2$ concentrations are in the range of previous studies performed on various environmental conditions (Deguillaume et al., 2014; Marinoni et al., 2004; Valverde-Canossa et al., 2005).

We have assessed the correlation between hydrogen peroxide and the non-sea salt of sulphate fraction. A good determination

coefficient of $r^2 = 0.60$ was obtained indicating possible relationship between $H_2O_2$ and sulphate aqueous production. Additionally, a positive moderate correlation ($r^2 = 0.53$) was observed between $H_2O_2$ and total dissolved organic carbon. This relationship could be used as a proxy for the gas-phase VOC concentration and gas phase production of $H_2O_2$, since it can be probably produced at high $VOC/NO_x$ ratios as a result of peroxy radical termination reactions (Benedict et al., 2012).

Iron speciation is also evaluated (Figure S6). Soluble iron comes from its dissolution from crustal aerosols (low solubility) and from

anthropogenic and marine aerosols (high solubility). Its main redox forms are Fe(II) and Fe(III). This compound interacts with $H_xO_y$ compounds in cloud water through a complex redox cycle converting Fe(II) into Fe(III) and reciprocally (Deguillaume et al., 2005).



This leads to HO• production in cloud waters through for example the photolysis of Fe(III) and the Fenton reaction between Fe(II) and $H_2O_2$. It is expected during daytime conditions that Fe(III) is converted into Fe(II) due to efficient photochemical processes producing $HO_x$ radicals reacting with Fe(III) and due to Fe(III) photolysis. This is not systematically observed in our samples with, in average, an Fe(II)/(Fe(II)+Fe(III)) ratio equal to $52 \pm 22\%$. This suggests that the reduction of Fe(III) to Fe(II) is efficient (possibly due to for example the reaction with Cu) and possible complexation of Fe(III) with organic species, stabilizing iron under this redox form (Fomba et al., 2015; Parazols et al., 2006; Vinatier et al., 2016). Moreover, iron is not expected in those cloud samples to represent a significant source of HO• radicals due to low levels of concentration (less than 0.45 µM in average) (Bianco et al., 2015).

### 3.3 Dissolved organic matter in cloud water

To characterise the organic matter dissolved in cloud water, several targeted analyses were performed. These analyses allow the identification and quantification of amino acids, sugars, carboxylic acids, carbonyls compounds, and low-soluble VOCs. Those compounds have been selected for several reasons. First, they are supposed to be present in significant concentrations in the aqueous phase, based on previous studies. Second, they are known to be representative of sources (biogenic, marine, anthropogenic) or processes (chemical reactivity, biological activity) occurring within the atmosphere. Third, since the cloud medium is multiphasic, VOCs will help to evaluate the partitioning of the organic matter among the gas and liquid phases.

*Carboxylic acids*

Both short-chain mono- and di-carboxylic acids have been quantified in all the cloud water sampled at Reunion Island (Figure S7). Those compounds in the atmospheric aqueous phase present different sources. They come from their dissolution from the gas and particulate phases or result from chemical transformations from organic precursors in the aqueous phase (Rose et al., 2018). For example, the oxidative processing of aldehydes leads to the formation of carboxylic acids (Ervens et al., 2013; Franco et al., 2021) that can contribute to "aqSOA".

As expected, acetic and formic acids were the dominant species in all the cloud samples (average $63.9 \pm 63.8$ and $22.9 \pm 10.8$ µM, respectively). Lactic and oxalic acids (average $5.45 \pm 8.07$ and $0.79 \pm 0.55$ µM, respectively) are the other two main carboxylic acids. Oxalic acid is the smallest di-carboxylic acids with concentrations much lower than formic and acetic acids ranging from 0.27 to 1.78 µM. Finally, malonic, succinic, and malic acids account for ~70% of the less concentrated dicarboxylic acids fraction. Di-carboxylic acids represent in average 8.5% of the total measured carboxylic acids.

Acetic and formic acids can be emitted directly by anthropogenic and biogenic sources in gas or particulate phases and then diluted into the aqueous phase (Chebbi and Carlier, 1996), or they can be a result of chemical processing from organic oxygenated precursors (Charbouillot et al., 2012). Several in situ studies demonstrated that the source by mass transfer from the gas phase into the aqueous phase is important (Leriche et al., 2007; Sanhueza et al., 1992).

In the present study, a weak correlation ($r^2 = 0.26$) between formic and acetic acid is observed, indicating the contribution of different sources or formation pathways. The formic to acetic acid ratio (F/A) has been suggested to be a useful indicator of direct emission sources or secondary chemical formation in the aqueous phase (Fornaro and Gutz, 2003; Wang et al., 2011). F/A ratios lower than 1.0 indicate the contribution of primary anthropogenic emission, whereas photochemical oxidation of VOCs leads to higher concentrations of formic than acetic and, therefore an increase in F/A ratios (> 1.0) (Fornaro and Gutz, 2003). From our observations, a F/A ratio lower than 1 was obtained in most of the cloud events; with exception of the events R2 and R3 where the concentrations of formic were most abundant. This result suggests that primary emissions could regulate the concentrations of both carboxylic acids in the aqueous phase for most of the sampled clouds. For these two events, formic acid could also be produced by aqueous phase oxidation of aqueous formaldehyde that strongly depends on its concentration in the gas phase. The average concentrations of acetic and formic acids observed at Reunion Island were much higher than those observed in previous studies performed at sites under marine influence such as the East peak in Puerto Rico (by factors of 16.8 and 7.6 for marine clouds, respectively (Gioda et al., 2011)) or at the PUY observatory for clouds classified as "highly marine" (by factors of 4.64 for acetic but similar levels of formic acid (Renard et al., 2020)). Our values are also higher than measurements performed at the Schmücke mountain in Germany (4.17 and





1.5 times higher for minimum average values, (van Pinxteren et al., 2005)), or at the Tai Mo Shan in Hong Kong (by factors of 6.69 and 1.34, (Li et al., 2020)). These discrepancies when compared with other studies could be related to the influence of direct primary biogenic sources, such as vegetation and soil emissions (Talbot et al., 1990) and anthropogenic sources (biomass burning and/or vehicular emissions, (Rosado-Reyes and Francisco, 2006; Talbot, 1995). Differences might be also related to the location of our sampling point in the tropical latitudes where photochemical processes of biogenic emissions could be enhanced leading to formic

production in the particle-aqueous phase (Liu et al., 2012b).

Di-carboxylic acids such as oxalate result mainly from particle dissolution and from aqueous phase reactivity (Charbouillot et al., 2012; Perri et al., 2009; Renard et al., 2015). Oxalic acid can be also produced by oligomerisation from biogenic products, such terpenes and isoprene under deliquescent condition (Renard et al., 2015). Oxalate displays lower average concentrations than formic and acetic acids ($0.79 \pm 0.56\,\mu M$). These values are consistent with study from Gioda et al. (2011) who found concentrations for a

marine site equal to $0.7 \pm 0.3\,\mu M$. These concentrations are lower than those observed at mid-latitude mountain sites such as the PUY observatory ($2.97 \pm 2.61\,\mu M$ for highly marine clouds (Renard et al., 2020)), at Mount Lu in South China ($4.95\,\mu M$ (Sun et al., 2016)), and at the Schmücke mountain in Germany ($1.7\text{-}2.1\,\mu M$ ranges (van Pinxteren et al., 2005)). As discussed in next sections, biogenic VOCs were observed in our study, suggesting the presence of freshly formed clouds, which could impact in the oligomerisation and secondary formation of oxalate in the liquid and condensate phase. However, since cloud measurements were

performed close to the emission sources, probably there was not enough time to observe the oxidation effects of biogenic products to oxalic acid, also considering that the concentrations of $H_2O_2$ were quite low. Malonic and succinic acids can be produced by photooxidation in the aqueous phase (Charbouillot et al., 2012). Additionally, succinic acid can be produced in marine environments by photochemical degradation of fatty acids at the sea microlayer (Kerminen et al., 2000). Malonic and succinic acids present similar or higher average concentrations in our study ($1.07 \pm 0.54$ and $0.94 \pm 0.34\,\mu M$, respectively) than at PUY ($0.59$ and $0.66\,\mu M$ (Renard

et al., 2020)), and at the Schmücke mountain ($0.3\text{-}0.4$, $0.3\text{-}0.5\,\mu M$ (van Pinxteren et al., 2005)). Future investigations using explicit cloud chemistry models could help to better understand the formation pathways of mono and di-carboxylic acids in this tropical environment and to evaluate the contribution of aqueous phase reactivity on the chemical budget of carboxylic acids and carbonyls.

*Amino Acids*

In the last years, several studies have been carried out to analyse free or combined amino acids (AAs) in the atmosphere, for various

environmental sites. Their atmospheric interest relies on the fact that AAs play a potential role on cloud formation processes (Kristensson et al., 2010; Li et al., 2013) as well as on atmospheric chemistry by reacting with atmospheric oxidants (Bianco et al., 2016). AAs also significantly contribute to organic nitrogen and carbon in atmospheric depositions and are sources of nutrients to various ecosystems. They are also known to be chemical tracers of different sources (terrestrial, biogenic, *etc.*) (Barbaro et al., 2011). AAs have been characterized in different atmospheric matrices such as aerosols (Mashayekhy Rad et al., 2019; Matos et al., 2016;

Ruiz-Jimenez et al., 2021; Scalabrin et al., 2012), rain waters (Xu et al., 2019), fogs (Zhang and Anastasio, 2003) and more recently in cloud waters (Bianco et al., 2016; Renard et al., 2021; Triesch et al., 2021). However, due to the challenges related to the cloud water sampling, little is known about the distribution and concentrations of AAs in contrasted environments.

Here we discuss the results of 15 AAs detected in the 12 cloud events at Reunion Island. The AA concentrations measured are represented in Figure 4. A significant variability in the total AA concentrations (TCAA) is observed between the cloud events,

ranging from 0.81 to 21.09 $\mu M$ (66.93 to 947.6 $\mu gC\,L^{-1}$). Similar range of concentrations has been observed at PUY (remote site under marine influence, France) by Bianco et al. (2016) (TCAA of 1.30 to 6.25 $\mu M$ ($211 \pm 12\,\mu gC\,L^{-1}$)) and by Renard et al. (2021) (TCAA of 1.06 to 7.86 $\mu M$ ($123.97 \pm 98.77\,\mu gC\,L^{-1}$)). In addition, a characterization of AAs was performed in cloud water samples collected at a marine site in Cape Verde (Triesch et al., 2021). These results also showed a remarkable variability of concentrations with values varying from 17 to 757 $\mu gC\,L^{-1}$. The differences in AAs concentrations between our study and those performed at PUY

and Cape Verde could be associated with the location of the sampling points. Although all sites were remote and under the influence of marine sources, our sampling site was surrounded by biogenic sources.





Regarding the AAs distribution, Ser, Ala and Gly dominate the total average concentration in our observations by 25, 18 and 13.6%, respectively (Figure 4). The same three AAs, together with Asn and Ile/Leu, were also dominant in cloud water samples collected at PUY (Renard et al., 2021) and were highly present in those collected at Cabo Verde (Triesch et al., 2021). This can be explained by their lower reactivity with HO• radicals, $O_3$ and $^1O_2$ in atmospheric waters leading to higher mean lifetimes as described from theoretical calculations (Jaber et al., 2021a; McGregor and Anastasio, 2001a; Triesch et al., 2021) and higher atmospheric concentrations. In addition, experimental work performed in microcosms mimicking cloud environment showed that the biodegradation and photo degradation rates of Ser were low; Gly could be also bio- and photo-produced under these conditions (Jaber et al., 2021b). Previous studies on aerosols have associated Gly and Ala with long-range marine aerosol transport (Barbaro et al., 2015; Mashayekhy Rad et al., 2019; Scalabrin et al., 2012) and Ser with terrestrial and marine aerosols (Mashayekhy Rad et al., 2019) and primary marine production (Scalabrin et al., 2012). However, their relative contribution is not constant throughout the field campaign. Ser and Ala are highly present all along the campaign, reaching up to 60% to the total AA's contribution, potentially suggesting the strong influence of long-range and local transport of marine aerosols. Gly represents a considerable AA fraction (up to 15%), but it is only observed during six cloud events, mostly observed in the last days of the campaign. A large concentration variability was also observed for Gly and Ser in cloud collected at PUY (Renard et al., 2021), and for Ala at Cabo Verde (Triesch et al., 2021). Tyr presents an opposite trend with maximum contribution (up to 39%) mostly during the first days. A similar pattern is also observed for Trp but with lower concentration. Met and Trp have the lowest concentrations similarly to clouds at PUY (Renard et al., 2021) and Cabo Verde (Triesch et al., 2021) these low concentrations are consistent with their short residence time calculated from their high reactivity with HO•, $O_3$ and $^1O_2$ (Jaber et al., 2021b; McGregor and Anastasio, 2001b; Triesch et al., 2021). The case of Tyr is more complex as its relative concentration is higher than those of Trp and Met. Although theoretical calculations predict that Tyr is highly reactive with atmospheric oxidant and should be in very low concentration, experimental work shows a relatively slow photo- and biodegradation of this AAs. Tyr has been found and associated to terrestrial aerosols, indicating a combination of various sources such as plants, bacteria and pollen (Scalabrin et al., 2012). High variability among the samples in the AAs distribution were also observed in other studies at PUY and Cape Verde. Nevertheless, globally major AAs reported by the literature are in agreement with our observations (Renard et al., 2021; Triesch et al., 2021). A complementary analysis about AAs sources is discussed in the section 4.2.

***Anhydro sugars, polyols and saccharides***

The speciation of anhydro sugars, sugar alcohols (polyols), and primary saccharides are analysed in this study. This is the first time to our knowledge that cloud water sugar composition has been investigated. The last two families are ubiquitous in the water-soluble fraction of atmospheric PM (Gosselin et al., 2016) and come from biologically derived sources (Verma et al., 2018). Some studies used those compounds as marker compounds to characterize and apportion primary biogenic organic aerosols (PBOAs) in the atmospheric particulate fraction (Samaké et al., 2019b, 2019a). PBOAs integrate bacterial and fungal cells or spores, viruses, or microbial fragments such as endotoxins and mycotoxins as well as pollens and plant debris (Amato et al., 2017; Deguillaume et al., 2008). Anhydro sugars are specific tracers of different sources: levoglucosan and its isomers are indicators of biomass burning (Simoneit et al., 1999). Glucose is representative of plant material (pollens, plant debris) (Pietrogrande et al., 2014); arabitol and mannitol are sugar alcohols associated with fungi emissions (Gosselin et al., 2016); trehalose is a metabolite of microorganisms and is suggested to be an indicator of the soil microflora (Jia et al., 2010). However, the full processes and sources associated to the presence of sugars in the atmosphere are still not fully known. Figure 5 reports the sugar concentrations identified in each cloud event and the average relative contribution for all the campaign. Sorbitol, glycerol, glucose, and mannitol were the most abundant species observed in the cloud samples. Globally, similar total sugar concentrations were found in the cloud events, with exception of R11, which presented higher total concentrations by a factor of 2 (Figure 5). Even though sugars and polyols can be produced in the atmosphere because of microorganisms' activity in the aqueous phase (production and consumption), their presence in cloud water can also result from their transfer from the aerosol phase. The sugar's profile observed in cloud water at Reunion Island is quite dissimilar to that observed in aerosols measurements, where glucose, mannitol and arabitol present the most abundant


concentrations (Samaké et al., 2019b). Thus, this result suggests the presence of other sources rather than aerosols, but their characterisation need to be further investigated.

### *Carbonyls (OVOC) and low-soluble VOC*

Concentrations of seven carbonyl compounds are obtained for eleven cloud samples. Total average carbonyl concentrations observed in cloud events ranged from 30.64 to 146.1 µgC L$^{-1}$ (Figure 6). Globally, formaldehyde (F, 1.40 ± 0.68 µM) presents the highest

average concentrations followed by hydroxyacetaldehyde (HyAC, 0.54 ± 0.46 µM), acetaldehyde (A, 0.48 ± 0.87 µM) and glyoxal (GL, 0.37 ± 0.56 µM). Average carbonyl concentration (75.2 ± 36.2 µgC L$^{-1}$, 3.5 ± 1.7 µM) is similar to that observed for highly marine clouds at PUY station, France (67 µgC L$^{-1}$, Deguillaume et al., 2014). These values differ from the previous observations under various environmental conditions: at Great lakes region (12.3 µM median concentrations, (Li et al., 2008)) Schmücke, Germany (3.8-10.3 µM, (van Pinxteren et al., 2005)), Hong Kong, China (19.4-74 µM, (Li et al., 2020)), and Whistler, Canada (12-

25 µM, (Ervens et al., 2013)). For all those studies, average carbonyls concentrations are higher. These dissimilar levels could be associated with the differences in carbonyls precursors and to the availability of oxidants at different locations. The tropospheric removal processes for carbonyl compounds in the gas phase are photolysis and reaction with the HO• radicals (Atkinson, 2000), but wet scavenging is also an important sink of carbonyls in the atmosphere. Indeed, carbonyls in cloud water are predominantly an outcome of their gas phase dissolution into the aqueous phase as a result of their Henry's law constants (Deguillaume et al., 2014;

Matsumoto et al., 2005). Levels of carbonyls in the aqueous phase will therefore strongly depend on their levels in the gaseous phase and on the solubility in the cloud water. Henry's law constants are quite dissimilar between carbonyl compounds, ranging from 3.2 10$^3$ to 9.9 10$^5$ M atm$^{-1}$ for F and GL, respectively, being H(GL)>H(HyAC)>H(F)>H(AC). The discussion about the sources of carbonyls in the cloud water is further discussed in section 4.1.

As observed with other organic species, a relevant variability in terms of total concentration and speciation is observed between the

cloud waters. To go further, the formaldehyde/acetaldehyde ratio (F/A) has been evaluated since it is widely used as an indicator to investigate the potential sources of carbonyls. A higher F/A ratio (up to 10) is usually observed in rural or near forest areas due to the fact that biogenic VOCs (such as isoprene) produce more formaldehyde than acetaldehyde through photochemical reactions (Shepson et al., 1991). In contrast, the F/A ratios in areas under the influence of anthropogenic emissions are much lower (generally lower than 2), attributed to the large amount of anthropogenic hydrocarbons being release (Shepson et al., 1991). In our study, the

average F/A ratio obtained for all the cloud samples was 6.7 ± 3.3, suggesting the contribution from vegetation emissions and similar to this observed in previous studies near rural areas (Yang et al., 2017). Even though, most of the samples show a F/A ratio ranged from 2.62-11.6, the event R1 presents a dissimilar ratio of 0.43 suggesting a plausible contribution of anthropogenic emissions for this specific case.

Only few studies have reported the presence of low-soluble VOCs in cloud water or fog droplets (effective Henry's law constant

between 1.82 10$^{-1}$ and 4.76 10$^{-3}$ M atm$^{-1}$). The compounds are from biogenic (mainly terpenoids and isoprene) and from anthropogenic (mainly aromatics) sources. Even these compounds present low concentrations in cloud waters, they have been targeted because (1) they are representative of distinct sources, (2) they present sanitary effect and are transferred to the ground by rain, potentially impacting other ecosystems. For instance, measurements performed at the remote Gibbs peak (USA) in cloud waters have shown average concentrations of ethylbenzene, o-xylene and toluene ~ 1.6, 4.2, and 6.5 nM (Aneja, 1993). A recent work

reports the concentration of these compounds at PUY, presenting similar average values than those observed at Gibbs Peak (1.88-4.7 nM, Wang et al., 2020). In this study, nine different volatile organic compounds were identified in the cloud samples applying the technique developed at PUY by Wang and co-workers (2020). They are mostly terpenoids (α-pinene, β- pinene and limonene) and isoprene which are released by terrestrial vegetation, like conifers and deciduous trees (Fuentes et al., 2000). They are also primary aromatics (benzene, toluene, ethylbenzene, xylenes) usually related to fossil fuel combustion and evaporation emissions as

well as solvent use related activities (Borbon et al., 2018). These compounds are known to be present in the gas phase and then dissolved into the aqueous phase.





Isoprene, α-pinene and β-pinene depict the higher concentrations, with average values of 19.37 ± 6.75, 7.99 ± 21.16 and 7.45 ± 14.51 nM, respectively. Aromatics present lower concentrations, ranging between 0.29 ± 0.21 to 2.15 ± 2.02 nM. Concentrations of biogenic compounds surpass the PUY concentrations by factors of 2 to 10. Concentrations of aromatics are lower than those observed at PUY by factors of 36 for toluene, 16 for o-xylene and 3 for benzene.

As depicted in Figure 6, biogenic compounds dominate the VOC fraction at Reunion Island, suggesting the influence of emission from vegetation due to the presence of the nearby endogenous "tamarin forest". A recent study has shown that cloud processing of isoprene products is responsible for the 20% of the total biogenic SOA burden (Lamkaddam et al., 2021). These results raise the question about the role of terpenoids and their oxidation products in the aqSOA formation and should be further investigated.

*Characterisation of DOC from targeted organic analyses*

Figure7a shows the total dissolved organic carbon (DOC) observed in cloud water samples. A significant variability between events can be observed, ranging from 5.82 to 62.0 mgC L$^{-1}$, with average values of 25.46 ± 19.20 mgC L$^{-1}$. Herckes et al. (2013) reported in his review the various sites where TOC/DOC were measured all over the world. The TOC levels in this study are significantly higher (by a factor of 5) than those observed at PUY for highly marine clouds (Deguillaume et al., 2014). The study from Benedict et al. (2012) performed for marine cloud waters sampled at the southern Pacific Ocean shows DOC concentrations ranging from 1.32 to 3.48 mgC L$^{-1}$. Even lower TOC concentrations were reported in Puerto Rico for marine clouds which ranged from 0.15 to 0.66 mgC L$^{-1}$ (Reyes-Rodríguez et al., 2009). The DOC levels observed at Reunion Island are surprisingly high for a site under a marine influence, suggesting the contribution of important sources of DOC other than sea-related ones.

Figure 7b reports the relative characterised organic fraction associated to the DOC measured for each cloud event. Only the samples with full targeted analysis were considered here. As expected, the total characterised organic fraction is strongly variable since it depends on the dissolved organic carbon in cloud waters. On average, 20% of the organic composition is identified during our campaign reaching up to 35% in some cases. Carboxylic acids and sugars depicted the maximum contribution, representing 10.72% and 7.11%, respectively. Amino acids only represent a 1.64% contribution on average. Previous studies have shown a higher contribution of AAs to the TOC, reaching up to 10% in marine clouds (Bianco et al., 2016). This difference could be explained by the heavier dominant AA species measured in Bianco et al. (2016) (Trp, Phe, Ile) compared to our study (Ala, Gly and Ser) and by the lowest TOC amount at PUY for this specific study. Carbonyls and low-soluble VOCs represent the lower fraction of the total dissolved organic carbon, reaching up to 1% for OVOCs and 0.35% for VOCs to the total. Even though an extensive analysis of the dissolved organic matter is evaluated here, our results shown that there is still a considerable fraction of OM not characterized. Further non-targeted analysis (future work) might be useful to better understand the sources and processes involved in the contribution of organic carbon in tropical-marine cloud water.

## 4. Discussion

### 4.1 Gas-aqueous phase partitioning of OVOCs and low-soluble VOCs

During the BIO-MAÏDO campaign, OVOCs and low-soluble VOCs were simultaneously captured on Tenax tube during the cloud sampling (see section 2 on the AEROVOCC gaseous sampler) and the gaseous concentrations were evaluated by TD-GC-MS analysis. A recent study has shown the potentiality of the on-sorbent pre-coated PFBHA tube method (Rossignol et al., 2012). However, our tests described in the supplement material raised the critical issues associated to breakthrough volume for light OVOCs. Since the breakthrough volume limits the observed gaseous concentrations, our calculation is semi-quantitative and the OVOC levels discuss here should be considered as a lower limit (and an upper limit for partition calculations). Only few previous studies have analysed the partition of these compounds between gas and aqueous phases. Thus, despite the technique's limitations for OVOCs we have included their analysis as they could be key compounds to better understand the multiphasic cloud chemistry. OVOC and VOC concentrations obtained are presented in Table S7. This allows the evaluation of their partitioning between different atmospheric phases.





The most abundant gas-phase OVOCs are F, AC, methacrolein, butanal and hexanal (90%), while MGL (0.6%) and GL (0.7%) are the minor species. F represents between 38 and 93% of the total detected carbonyls with values ranging between 0.73 and 13.6 ppb.

Similarly, F dominates the carbonyl concentrations in the cloud phase. The lower concentrations of gaseous GL and MGL compared with those of formaldehyde could be associated to the oxidation yields of VOC precursors such as aromatics and terpenes or to its primary emissions (Friedfeld et al., 2002; Wolfe et al., 2016). Previous studies at Reunion Island have shown that gaseous formaldehyde is mainly associated to secondary biogenic products (37%) and primary anthropogenic emissions (14%) (Duflot et al., 2019; Rocco et al., 2020).

Low-soluble volatile compounds that have been quantified in the gas phase are from both biogenic and anthropogenic sources. Isoprene is the most abundant biogenic species with an average mixing ratio of 109.2 ± 77.63 ppt during the cloud events. Other biogenic compounds (pinenes and limonene) present lower concentrations between 4.93 and 35.8 ppt. Duflot et al. (2019) investigated the isoprene concentration over several sites at Reunion Island. An average concentration of 95 ppt was observed, which shows the same order of magnitude than our study (109.2 ppt, Table S7). In the Amazon forest, the concentration level of isoprene

(<500 ppt) is found higher probably due to the higher emission by the vegetation (Yáñez-Serrano et al., 2015). They have also observed pinenes and limonene. The sum of concentration levels of these compounds is less than 600 ppt. This is almost twice as high as the sum of pinenes and limonene concentrations during BIO-MAÏDO. Aromatic compounds have also been detected in the gas-phase during the cloud events with benzene and toluene presenting the highest levels (233.6 ± 223.4 and 220.2 ± 129.3 ppt, respectively). These concentrations are rather low and are representative of remote environment (Wang et al., 2020).

To investigate the partitioning of those compounds detected in parallel in both the gas and the aqueous phases, q factors have been calculated for each cloud events et for each compound following Wang et al.(2020). The q factor represents the deviation from the theoretical Henry's law equilibrium of the aqueous phase concentration of chemical species. The q factors superior to 1 reveals super-saturation in the aqueous phase and reciprocally. Many factors can explain these deviations such as sampling artefacts, chemical reactivity in both phases, kinetic limitations through the air/droplet surface. The evaluation of q factors requires the liquid

water content (LWC in vol of water / vol of air), the temperature, the effective Henry's law constants at measured temperature. q is calculated for each compound (Audiffren et al., 1998). All those data and the calculations are presented in Table S7 and Table S8. Results highlight small deviations from the equilibrium considering the Henry's law constant for OVOC (Figure 8). Due to the breakthrough volume issue, the reader should be aware that all the q partitioning coefficients are overestimated and have to be considered with caution. Formaldehyde is slightly subsaturated in the aqueous phase (average q factor around 0.12). Previous in situ

studies emphasized that formaldehyde partitioning is governed by Henry's law equilibrium (Li et al., 2008; van Pinxteren et al., 2005) or is slightly subsaturated (Ricci et al., 1998). One explanation is relative to kinetic transport limitations through the surface of the cloud droplets that can lead to small sub saturation of species coming only from their dissolution from the gas to the aqueous phase (Ervens, 2015). This can also be a reason leading to the small sub saturation of glyoxal in the aqueous phase (average q factor around 0.08) that is highly soluble ($H_{eff}$ between 8.0 to 9.9 $10^5$ M atm$^{-1}$, depending on the temperature). On the opposite,

methylglyoxal and acetone partitioning's exhibit small supersaturation (average q factor equal to 23 and 10, respectively). Supersaturations of these two compounds has been also observed in van Pinxteren et al. (2005) during the FEBUKO campaign in Germany. Low-soluble biogenic and anthropogenic VOCs present much higher super saturation in the aqueous phase as already observed at PUY by Wang et al. (2020). We observed super-saturations in the aqueous phase of those compounds with q factors varying between 2.63 $10^1$ to 1.16 $10^5$ (Figure 8).

Figure 9 represents the mean q factors for all the studied VOCs classified as a function of their effective Henry's law constant. We clearly observe that this factor becomes higher when the VOC solubility becomes lower. This super-saturation for hydrophobic compounds has been reported by Wang et al. (2020) in cloud water and by Glotfelty et al. (1987) and Valsaraj et al. (1993) in fog water. Many reasons have been highlighted to explain this statement such as the possible interactions with dissolved or colloidal matter or adsorption of organic species at the air-water interface (Valsaraj et al., 1993; Wang et al., 2020). This could be possibly





important in this context with the elevated level of TOC in the cloud samples. However, reasons for the observed deviations are not
       fully clear and can result from both physical and chemical effects.

**4.2 Environmental variability**

       This section is related to the analysis of the environmental variability of the chemical composition of clouds. It is important to
       highlight that this statistical analysis is limited by the few numbers of samples obtained during the field campaign. As previously
discussed, a strong chemical variability was observed between the cloud samples obtained in our study, mainly for organic
       compounds such as organic acids, carbonyls and amino acids. The presence of marine sources is clearly observed in the ionic
       composition of the samples, which dominate the inorganic fraction. Even though the contribution of other sources is observed, as
       noted for sulphate, their emission sources, and their role in the chemical cloud processing is not fully understood.

       The purpose of this evaluation is then a better understanding of the interactions between microphysics, chemistry, and dynamics (air
mass history) affecting the burden and fate of cloud chemistry in a tropical island.

*Influence of Air Mass History during BIO-MAÏDO campaign*

       In this section we provide and analyse the correlation between the concentration of the inorganic ions ($Na^+$, $NH_4^+$, $K^+$, $Mg^{2+}$, $Ca^{2+}$,
       $Cl^-$, $NO_3^-$ and $SO_4^{2-}$), the concentration of TOC, sugars, amino-acids and organic acids and microphysical parameter (LWC: liquid
water content), on the one hand, and their mass history on the other. During their atmospheric transports, the air masses receive
       chemical species under various forms (gases and particles) from various sources. This strongly depends on the altitude of the air
       masses. The results of the Mann-Whitney test for the data of 1000 and 500m above sea or ground level, do not show any significative
       difference. Thus, in this work we define that the air mass is potentially impacted by local emissions when transported in the layer
       500m above sea or ground level. This assumption is clearly questionable and sensitivity tests have been performed. During the
transport, chemicals could also undergo multiphasic chemical transformations, as well as dry or wet deposition. The objective, here,
       is to evaluate the effect of the history of air masses on the chemical composition of clouds.

       To this end, partial least square (PLS) regressions are performed to establish the correlations between the chemical categories,
       microphysical parameters, and the land use cover. The matrix of the explanatory variables (the "Xs") is composed of the LWC
       matrix and the "land use cover" matrix provided by interpolation on the back-trajectory points (Section 2.4, Figure 10). Percentages
in Table S9 have been calculated considering back trajectory points lower than 500 m above sea or ground level and then correlated
       with the 4 main land cover categories in order to obtain the relative contribution of each area. The matrix of the dependent variables
       (the "Ys") gathers four groups of compounds (individual concentrations of inorganic ions, (di)carboxylic acids, amino acids and
       sugars).

       The index of the predictive quality of the model is slightly negative ($Q^2$ = -0.078 with one component) which means, first, most of
the ion concentrations are collinear, the model is not predictive. Indeed, cloud chemical composition can be modulated by many
       other parameters than the chosen explanatory variables, related to the air mass history calculated by the model. Indeed, the cloud
       chemical composition depends on local microphysics (Möller et al., 1996; Moore et al., 2004; Wieprecht et al., 2005), as well as
       proximity to sources (Collett et al., 1990; Gioda et al., 2013; Kim et al., 2006; Watanabe et al., 2001), biological activity (Bianco et
       al., 2019; Vaïtilingom et al., 2013; Wei et al., 2017), seasons (Bourcier et al., 2012; Fu et al., 2012; Guo et al., 2012; Shapiro et al.,
2007), and diurnal cycles (Kundu et al., 2010). In addition, the collinearity (Figure S9) between the average concentration of ions,
       does not allow the categorization of clouds as performed in Renard et al. (2020). For instance, marine category (with predominant
       $Na^+$ or $Cl^-$) or continental one (with predominant $NH_4^+$ or $NO_3^-$), cannot be proposed for the collected samples since at Reunion
       Island a "well missed" distribution of ions is observed. This suggests that either air masses had the same history, or, more likely, the
       presence of physical phenomena lead to the "homogenization" of air masses. Furthermore, on the scale of a small island the use of
a very fragmented cadastre, does not show any substantial improvement in the understanding of the variability of the chemical
       content of the collected cloud events.





Table S10 displays the correlation matrix of this PLS, and given the Q², the interpretation is rather tricky, but one main trend emerges. "Farming area" is correlated with chemistry, in particular with amino acids ($R_{mean} = 0.39$), sugars ($R_{mean} = 0.40$), and dicarboxylic acids ($R_{mean} = 0.39$). Note that these correlations are slightly overestimated due to the weak anti-correlation ($R_{LWC} = - 0.27$) between
"Farming area" and the LWC. This tendency clearly requires further investigations.

*Focus on sugars and amino acids variability*

To go further, a deeper evaluation about sugars and amino acids in cloud water is provide here since those compounds are surely linked to the emissions by the surface of the island. Moreover, they are also well-known compounds that serve as tracers of sources.
To assess the potential sources of polyols, primary and anhydro saccharides, a multiple correlation with inorganic ions was performed. Figure S10 shows the correlation matrix obtained combining the sugars, ions, and light carboxylic acid concentrations. Strong correlations are observed for glucose and most of polyol species with calcium, which could suggest the influence of dust sources (Samaké et al., 2019b).

Strong correlations are also observed between polyols (inositol, sorbitol, arabitol and mannitol) with nitrate and potassium,
suggesting the contribution from biomass burning sources (Li et al., 2003). Interestingly, levoglucosan, a well-known biomass burning tracer, does not show any correlation with any of these ions. Additionally, good correlations were also observed between inositol and ammonium, which could suggest the contribution from cultures developed on the island.

The negative correlations of polyols with oxalate could indicate their contribution from primary emissions instead of secondary processing in the atmosphere. Non correlations between polyols and anhydro sugars with sulphate could suggest the secondary
aqueous formation of sulphate compared to the primary emissions of sugars (Samaké et al., 2019a, 2019b).

In this final section, we analyse more specifically AAs as a function of their physico-chemical properties based on the "hydropathy" index as suggested by Pommie et al. (2004) and adapted by Scalabrin et al. (2012). This classifies the amino acids as hydrophilic (Asp, Glu, Asn, Lys, Gln, Arg, Tyr), hydrophobic (Ala, Leu, Ile, Phe) or neutral (Gly, Ser, Thr, His). Barbaro et al. (2015) have used this classification on aerosols in the Antarctic and they remarked that hydrophilic AAs were prevalent in locally produced marine
aerosol, while hydrophobic ones occurred in aerosol collected at the continental station.

Figure S11 displays the contribution of each group of amino acids observed during our cloud sampling. As observed in other studies (Renard et al., 2021; Triesch et al., 2021), the dominant fractions of AAs are neutral (38%) and hydrophilic (36%) whereas the hydrophobic contribution is lower (26%). However, the results show dissimilar behaviour between the first cloud events (R2-R7) mainly dominated by hydrophilic AAs, whereas an increase in the neutral AAs fraction is observed in the last collected clouds (R8-
R13). Similar strong variation in the proportion of AAs based on the physico-chemical properties has been already observed in a study performed at Cabo Verde islands by Triesch et al. (2021). In our study, this change in the AAs composition can be related with a change also observed in the altitude of the air masses arriving at Reunion Island during our measurements. The dynamical analysis by back trajectories shows the influence of lower altitude air masses from the west region during the second part of the campaign, particularly for R9, R10A/B, and R13 events (Figures 10 and S7). In this period, high concentration of Gly, Ser (neutral)
and Asp (hydrophilic) are observed suggesting the influence of biogenic marine sources linked to the activity of diatoms and zooplankton in seawater (Triesch et al., 2021). Higher contribution of Arg is also observed in the last days, which is associated with plant growth (Scheller, 2001). Our results indicate that AAs composition in cloud water at Reunion Island might be related to a mix of oceanic and vegetation sources, which their relative contribution is varying with the air mass history.

**5. Conclusions**

We present here an exhaustive chemical and microphysical characterisation of cloud water composition obtained in the framework of the French BIO-MAÏDO project. The measurements were performed during an intensive field campaign in March-April 2019 at Reunion Island. This is the first time that such a large chemical characterisation is performed in cloud waters.





Even though Reunion Island is located at tropical latitudes with summits higher than 2000m, the formation of clouds along the slope of this mountainous island is regulated by the horizontal wind shear at higher altitudes which could block the vertical development of clouds produced in the slope of the island. This may explain the low concentration of liquid water during the field measurements. Indeed, when compared with previous studies at PUY, the average LWC values present lower concentration by factors of 2 to 3 on average.

Our findings show the presence of compounds of marine origin in cloud water samples (*e.g.*, chlorate, sodium and amino acids) demonstrating an ocean–cloud exchange. However, the non-sea salt fraction of sulphate reaches up to 80%, indicating the presence of other additional sources.

A temporal variability is observed in the organic content of clouds, with DOC levels reaching up to 62 mgC L$^{-1}$, a quite large content when compared with previous cloud/fog studies. This variability was not similar for all the targeted organic compounds evaluated, suggesting the presence of dissimilar emission sources or production mechanisms.

The environmental analysis indicates the correlation of several inorganic and organic species with the vegetation and farming land use zones, which dominate over urban and coastal emission areas. The correlations observed between sugars and polyols with some inorganic tracers, suggest the contribution of vegetation on the chemical content of clouds. Additionally, high levels of biogenic non-soluble VOCs (such as isoprene and terpenes) were detected, indicating the cloud-terrestrial vegetation exchange. These findings also depict some information about the cloud formation, suggesting the presence of "fresh" clouds highly influenced by the

island local sources. Furthermore, the age of the clouds is somehow supported by the low oxalate levels observed during our field campaigns.

The evaluation of gas-phase to cloud partition reveal the supersaturation of non-soluble VOCs. Similar results have been found at PUY (France) and at the Schmücke mountain (Germany). However, our results depict even higher supersaturation of terpenoids, suggesting their importance in the aqueous phase chemistry in highly impacted tropical areas.

Despite our large organic content characterisation in cloud water, there is still a significant DOC mass (65 to 80%) to be elucidated. Further efforts need to be made to address the main sources and processes responsible for the high organic content in this tropical site. Future investigations are planned in the frame of the BIO-MAÏDO using High Resolution Mass Spectrometry techniques (HRMS) to assess the organic fraction of the cloud water. This complete dataset will also serve for future modelling work with the objective to elucidate the role of sources, microphysical processes, chemical and biological transformations, and dynamics (*i.e.*,

transport) on the chemical composition of clouds.

**Acknowledgements.**

This work was funded by the French National Research Agency (ANR) thanks to two programs: ANR-18-CE0-0013-01 and ANR-17-MPGA-0013, also performed in the framework of the CAP 20-25 Clermont Auvergne Project. M. Brissy is acknowledged for

his participation in the collection of cloud samples. The authors also thank the Institut National de l'Information Géographique et Forestière (IGN) for the provision of Corine Land Cover (https://www.data.gouv.fr/fr/datasets/corine-land-cover-occupation-des-sols-en-france/).

**Data availability.**

Data is available in the supplementary material, but further information can be obtained under request.

**Author's contribution.**

PAD prepared the manuscript, analysed the data and designed the figures, with contributions from all authors. LD, MV, JLB, AB, AC, SH, JLJ, TB, FB, AMD, ML, JMP, MR, MR, PT and AV provided measurements, chemical analysis and data processing of the comprehensive dataset used in this study. JLB, MR and PT performed the model simulations and the calculation of back-trajectories used in this study. PR reanalysed the amino acids by UPLC-HRMS and calculated the uncertainties, contributed to the data analysis

of measured chemical species by applying PLS method together with back trajectories. ML is the principal investigator of the BIO-



MAÏDO project who has designed the field campaign and contributed to scientific discussion. LD provided overall guidance to experimental setup and design of cloud measurements, along with data analysis and interpretation of results, crucial for the scientific discussion.






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





**Figure's Captions**

Figure 1. Location of Piste Omega and other instrumented sampling sites at Reunion Island. Photo of the mast (10 m height) deployed during the BIO-MAÏDO project integrating the cloud water collector, cloud droplet probe ("CDP probe"), meteorological station ("Meteo") and gas phase collection ("AEROVOCC").

Figure 2. Distribution of liquid water content (LWC) and effective diameter (Deff) of cloud droplets observed during each cloud event at Reunion Island. R1 to R13 refer to individual cloud samples. Lower and upper box boundaries represent the 25th and 75th

percentiles, respectively; line inside box shows median, lower and upper error lines depicts the 10th and 90th percentiles, respectively; filled black circles represent the data falling outside 10th and 90th percentiles.

Figure 3. a) Ion concentrations observed for each cloud event and relative contribution of each species to the total average mass. R1 to R13 refer to individual cloud samples. b) Concentration ratios of chloride to sodium and sulphate to sodium obtained in each cloud event. Dashed lines represent the reference values of seawater content (1.17 (orange line) and 0.06 (green line), respectively,

(Holland, 1978)).

Figure 4. Amino acids concentrations and total average relative contributions observed in cloud waters. R1 to R13 refer to individual cloud samples.

Figure 5. Concentrations and total average relative contributions of polyols, anhydro and monosaccharides observed in cloud waters. R1 to R13 refer to individual cloud samples.

Figure 6. Concentrations and total relative contributions of carbonyls (μM) and low-soluble VOCs (nM) species observed in cloud waters. Sum TMB represents the sum of 1,2,4-trimethylbenzene; 1,3,5-trimethylbenzene and 1,2,3-trimethylbenzene concentrations. R1 to R13 refer to individual cloud samples.

Figure 7. Total dissolved organic carbon and relative contribution of organic compounds to the total organic content, observed for each cloud event and the relative average of total contributions during the field campaign at Reunion Island. R1 to R13 refer to

individual cloud samples.

Figure 8. Partitioning coefficient q factors calculated for individual cloud samples and individual OVOC (on the left) and low-soluble VOC (on the right), considering the average temperature and LWC measured during the sampling of each cloud event. Average corresponds to the average q factors for the cloud events. Partition coefficients are calculated when measurements in the both aqueous and gas phases were available and validated. R1 to R13 refer to individual cloud samples. More details about the

samples can be found in tables S1 and S6.

Figure 9. OVOCs/VOCs q factors averaged for all the cloud events and classified as a function of their effective Henry's law constants. Each color represents a group of compounds, namely terpenoids (green), aromatics (red) and carbonyls (OVOCs, blue).

Figure 10. Average back-trajectories for each cloud event (R1 to R13) obtained by the MesoCAT model. Small dots represent the air mass position every 5 minutes and large dots every hour, limited by the model resolution. Trajectories are color-coded (minimum

and maximum of the air masses altitude) by the average altitude.



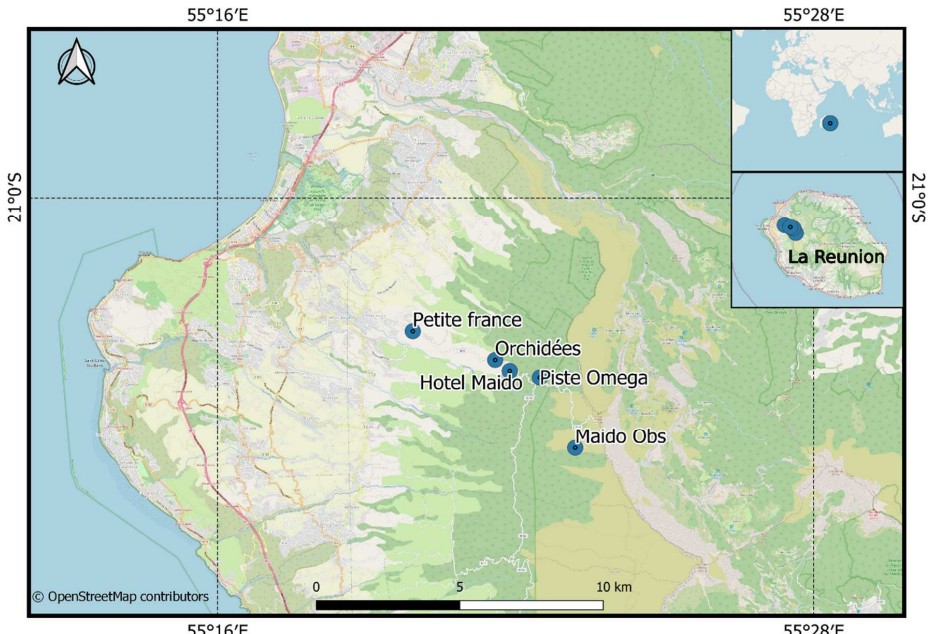

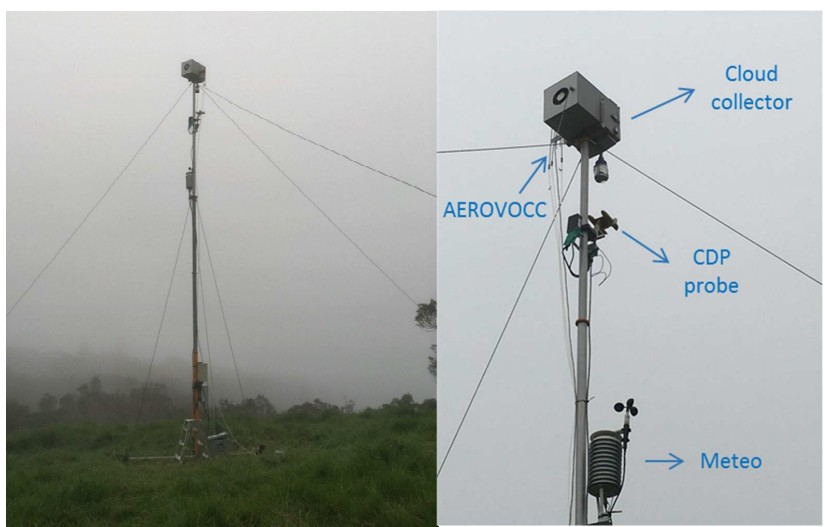

**Figure 1. Location of Piste Omega and other instrumented sampling sites at Reunion Island. Photo of the mast (10 m height) deployed during the BIO-MAÏDO project integrating the cloud water collector, cloud droplet probe ("CDP probe"), meteorological station ("Meteo") and gas phase collection ("AEROVOCC"). © OpenStreetMap contributors 2021. Distributed under the Open Data Commons Open Database License (ODbL) v1.0.**






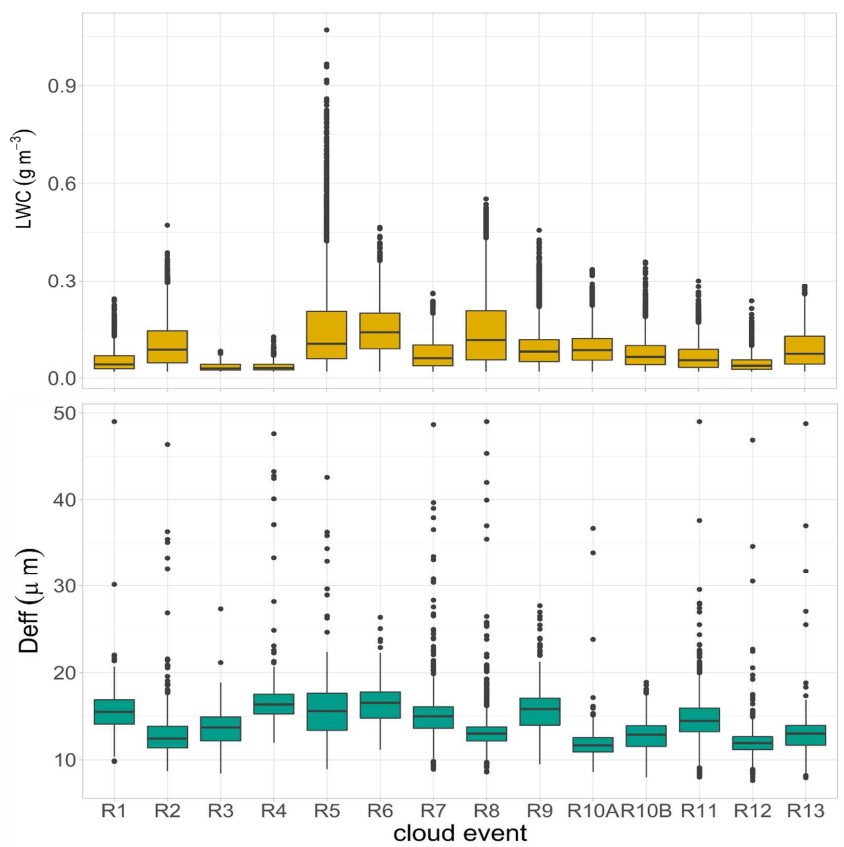

**Figure 2**. **Distribution of liquid water content (LWC) and effective diameter (Deff) of cloud droplets observed during each cloud event at Reunion Island. R1 to R13 refer to individual cloud samples. Lower and upper box boundaries represent the 25th and 75th percentiles, respectively; line inside box shows median, lower and upper error lines depicts the 10th and 90th percentiles, respectively; filled black circles represent the data falling outside 10th and 90th percentiles.**




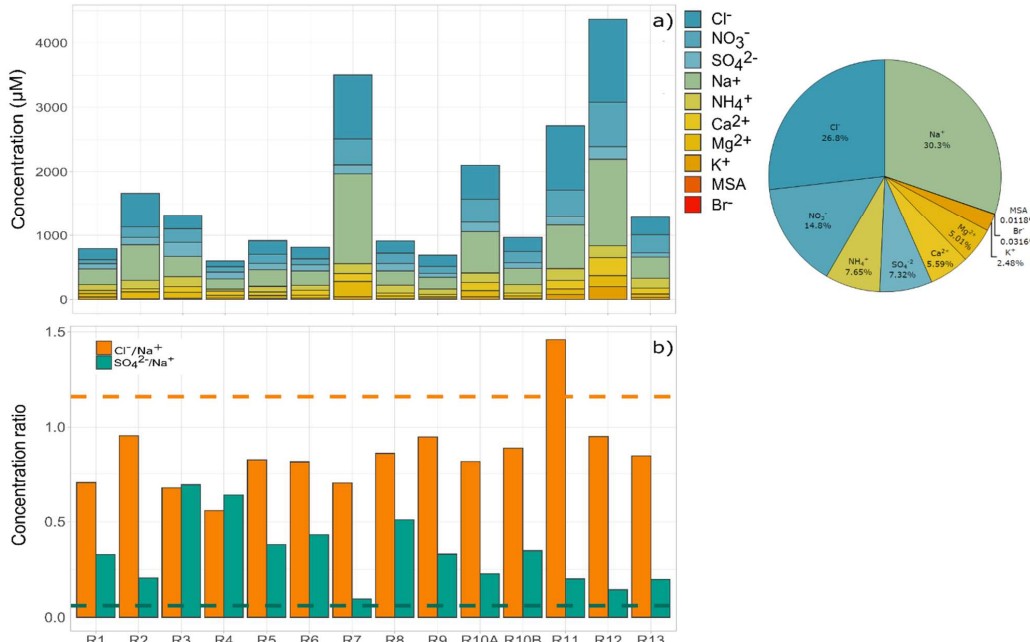

**Figure 3. a) Ion concentrations observed for each cloud event and relative contribution of each species to the total average mass. R1 to R13 refer to individual cloud samples. b) Concentration ratios of chloride to sodium and sulphate to sodium obtained in each cloud event. Dashed lines represent the reference values of seawater content (1.17 (orange line) and 0.06 (green line), respectively, (Holland, 1978)).**






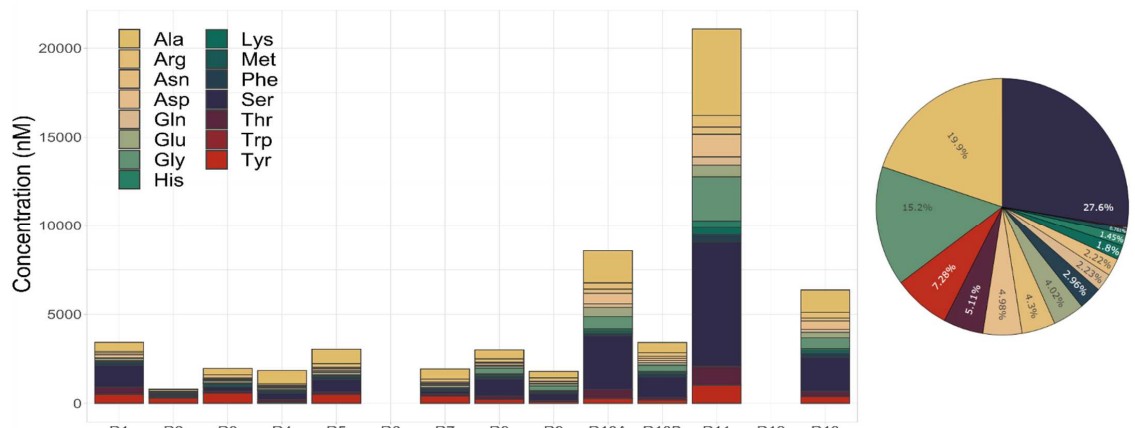

**Figure 4.** Amino acids concentrations and total average relative contributions observed in cloud waters. R1 to R13 refer to individual cloud samples.





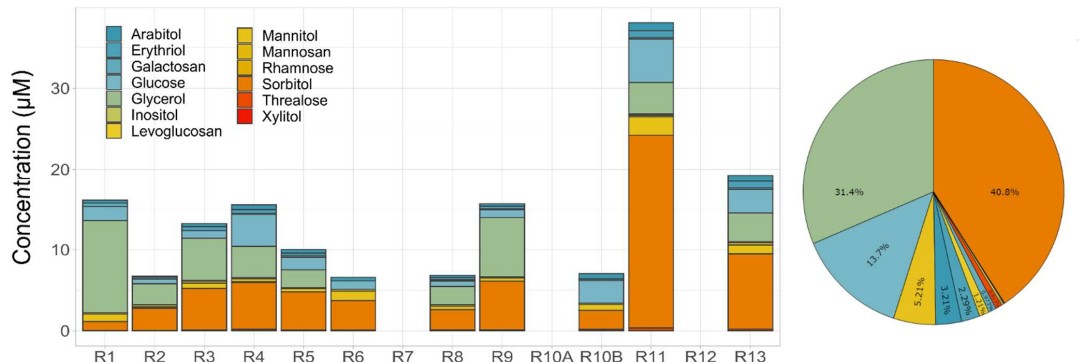


**Figure 5. Concentrations and total average relative contributions of polyols, anhydro and monosaccharides observed in cloud waters. R1 to R13 refer to individual cloud samples.**





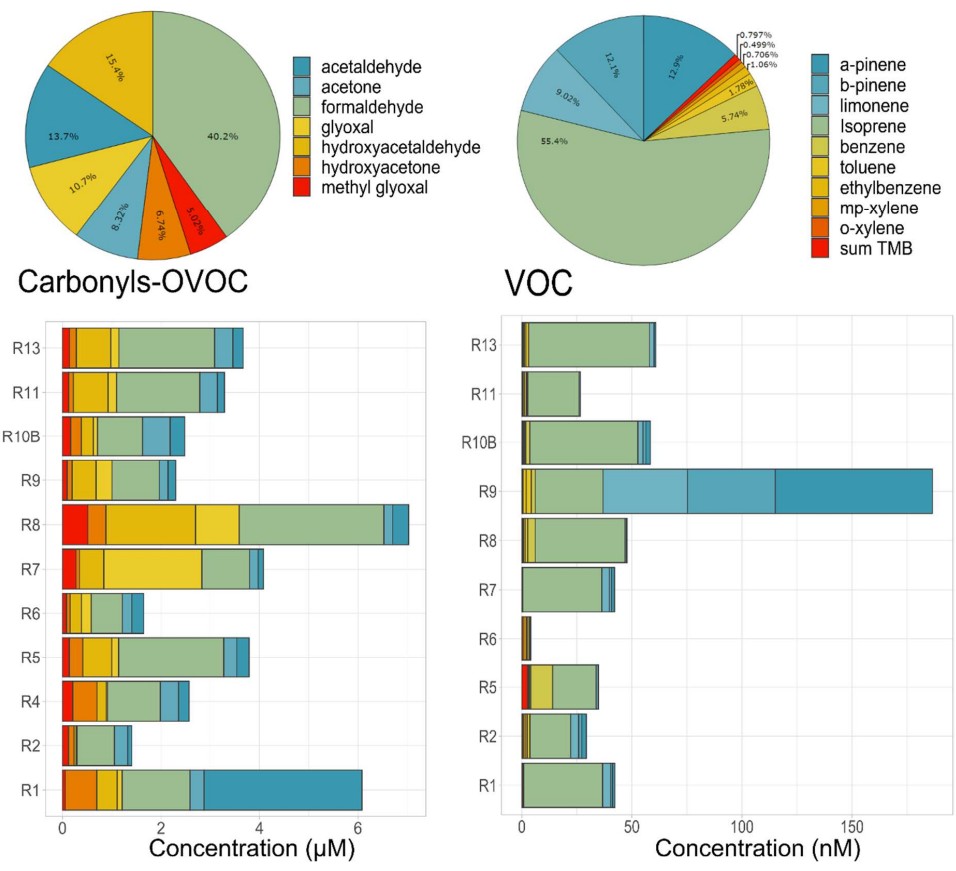


**Figure 6. Concentrations and total relative contributions of carbonyls (µM) and low-soluble VOCs (nM) species observed in cloud waters. Sum TMB represents the sum of 1,2,4-trimethylbenzene; 1,3,5-trimethylbenzene and 1,2,3-trimethylbenzene concentrations. R1 to R13 refer to individual cloud samples.**






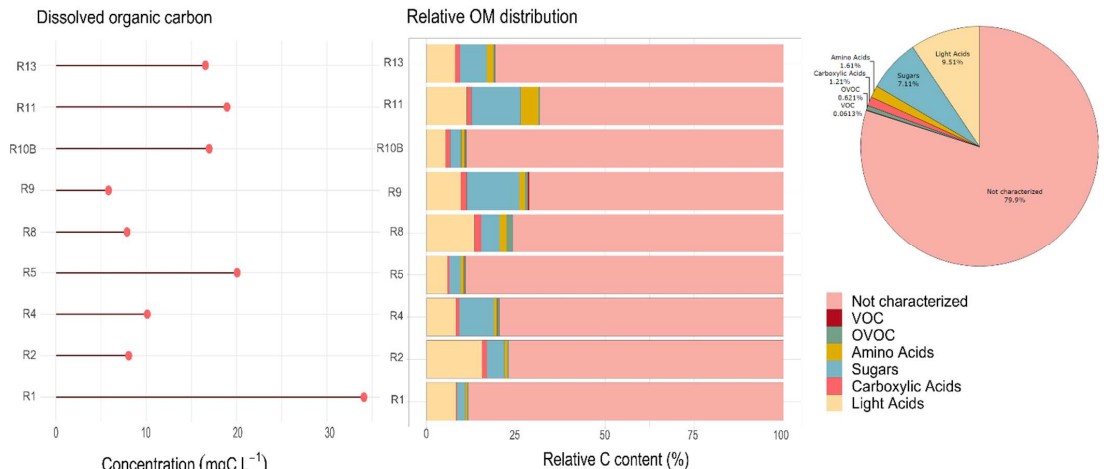

**Figure 7**. **Total dissolved organic carbon and relative contribution of organic compounds to the total organic content, observed for each cloud event and the relative average of total contributions during the field campaign at Reunion Island. R1 to R13 refer to individual cloud samples.**






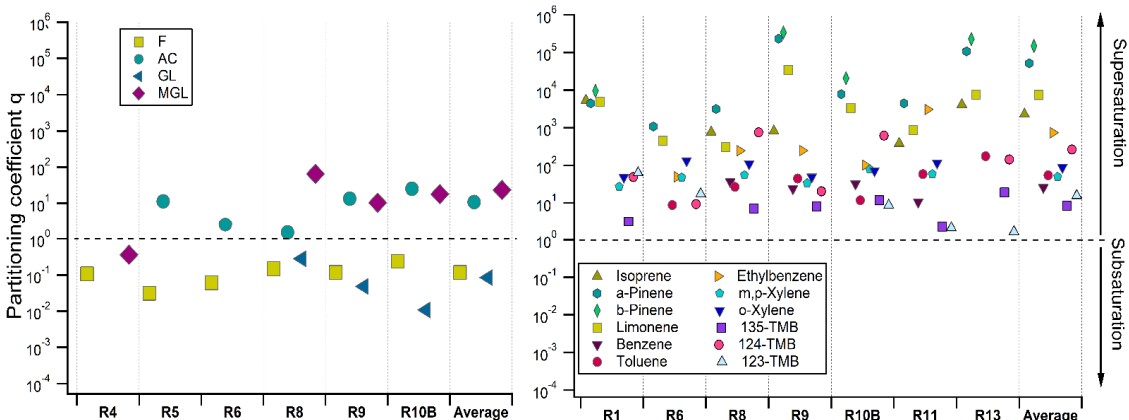

**Figure 8. Partitioning coefficient q factors calculated for individual cloud samples and individual OVOC (on the left) and low-soluble VOC (on the right), considering the average temperature and LWC measured during the sampling of each cloud event. Average corresponds to the average q factors for the cloud events. Partition coefficients are calculated when measurements in the both aqueous and gas phases were available and validated. R1 to R13 refer to individual cloud samples. More details about the samples can be found in tables S1 and S6.**

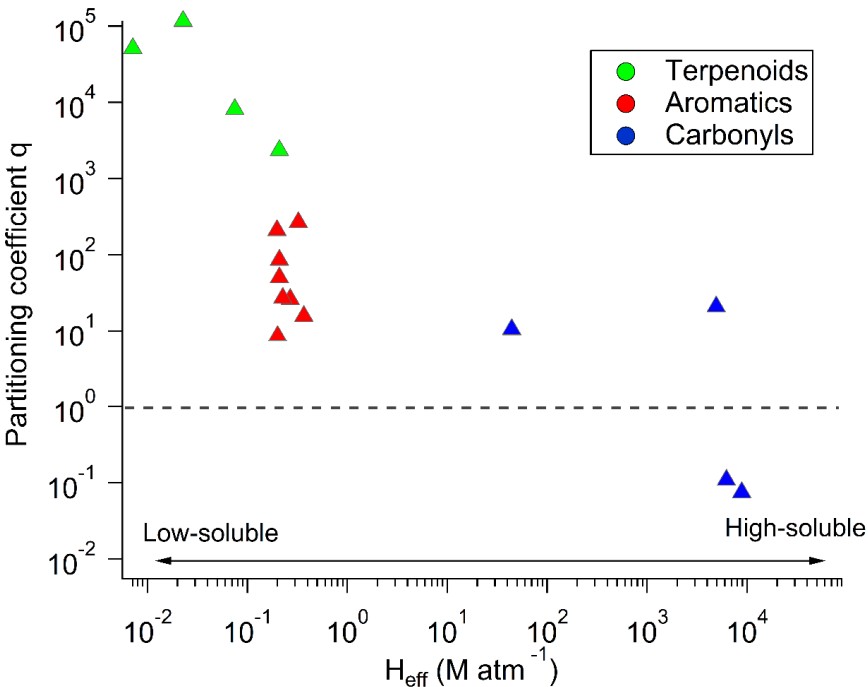

**Figure 9.** OVOCs/VOCs q factors averaged for all the cloud events and classified as a function of their effective Henry's law constants. Each color represents a group of compounds, namely terpenoids (green), aromatics (red) and carbonyls (OVOCs, blue).



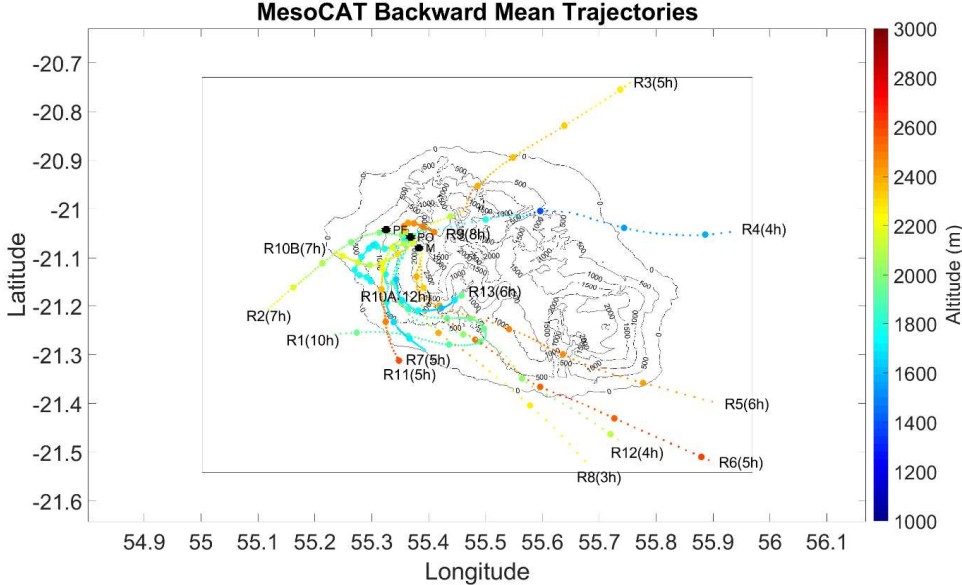

**Figure 10. Average back-trajectories for each cloud event (R1 to R13) obtained by the MesoCAT model. Small dots represent the air mass position every 5 minutes and large dots every hour, limited by the model resolution. Trajectories are color-coded (minimum and maximum of the air masses altitude) by the average altitude.**
