# Peer review of "Insights into tropical cloud chemistry at Reunion Island (Indian Ocean): results from the BIO-MAÏDO campaign"

_Atmospheric Chemistry and Physics, 2021_

## Author Comment (AC1)

**Author responses to referee comments of acp-2021-518**

Dominutti P. et al.

**Response to Reviewers**

The authors thank the reviewers for their positive and constructive comments relating to this paper. We address the specific points of each reviewer below. Reviewer comments are shown in black, author response in orange, and text added or amended in paper shown in blue.

**Referee 1**

The article deals with the physical-chemical characterization of cloud water samples collected in a pristine region. The article is very well written, with excellent figures and tables and an indepth discussion of the results. In addition, the authors made a detailed and careful description of the analytical methods used to determine the chemical composition, which makes the results reliable. Due to the degree of explanation, I have no doubts about the results.

We would like to thank the reviewer for their positive comments.

Just a few comments:

- How far is the island from the mainland or from the nearest anthropogenic sources?

The island is about 750 km far from Madagascar (Island) and approximately 2000 km far from the mainland (African continent).

- Why was the conductivity not measured?

Unfortunately, conductivity measurements were not performed during this field campaign.

- Have no preservative agents been added to the cloud water samples?

All the samples were filtered after collection to remove microorganisms. Samples were stored under frozen or refrigerated conditions based on storage procedures (see the SI1 section "sample conservation" for more information).

**- Did the cloud sampler blanks present any contamination?**

The cloud water collector was firstly cleaned with MilliQ water and let dry under a fume hood. Once dried, the three pieces were placed in sterilization bags with clean nitrile gloves and then sterilized to avoid microbiological contamination. The pieces were extracted from sterilization bags and mounted with sterile gloves immediately before sampling. Chemical and microbiological contaminations were tested at the beginning of the campaign, by spreading sterilized MilliQ water on the clean cloud collector and analyzing it with different approaches. This blank sample will therefore be named "blank" in the following text.

Since the cloud collector is made of Aluminium, we brought particular attention to metal contamination. Blank correction was taken into account for trace metal analysis, where the concentrations found in the blank were subtracted to the concentrations measured in the samples. Similarly, Fe(II) and Fe(III) concentrations were measured in the blank and subtracted

for the samples. The metal concentrations found in the blank were similar to the ones already measured for previous campaigns at the puy de Dôme station.

Analogously, the concentration of main inorganic and organic ions, measured by ion chromatography, resulted to be similar to the blank concentration measured previously at the puy de Dôme station and negligible compared to ions concentration in the samples. Microbiological contamination was tested by culture of 100  $\mu$ L of blank on R2A and no microbial colonies were observed after 7 days at 15°C.

A sentence was added in the text:

"Chemical and microbiological contaminations were tested at the beginning of the campaign, by spreading sterilized MilliQ water on the clean cloud collector and analyzing it with different approaches. Concentration of contaminants was subtracted to sample concentration in the case of trace metal analysis and Fe(II) and Fe(III) analysis, while it was considered to be negligible for main inorganic and organic ions and microbial concentration."

**- Have the first mL of cloud water been discarded?**

This is a good remark. For cloud collector such as CASCC (Caltech active strand cloudwater collector), this procedure is commonly performed to avoid contamination since the collection of droplets is performed using Teflon strands. However we use in this study a cloud impactor that is composed of aluminum plates to impact cloud droplets and collect them. Cloud droplets then fall down in a sterile bottle through a collection funnel also made in aluminum. All those parts of the collector are cleaned and sterilized before each sampling; so, we decided to not discard the first mL of cloud water.

Some expressions are in disuse in Analytical Chemistry:

- it is always necessary to separate the unit from the number 90 %
- M the correct one is mol L-1
- calibration curve the correct one is analytical curve

Thanks for these comments, the manuscript has been updated with these changes.

**Referee 2**

The manuscript presents and discusses results from chemical and physical characterization of clouds at Reunion Island. The study is impressively extensive with regards to the many chemical parameters measured in one study. The manuscript is very descriptive but the results are novel enough to justify publication as not many studies exist, especially in this kind of environment. The discussion is at times quite superficial though and does not lead to clear conclusions. Hence one wonders if some parts could possibly be omitted. Quite a few items should be clarified and potential artifacts excluded before publication of the manuscript.

**Major points**

The authors see a substantial amount of non-sea salt sulfate. Given that Reunion Island has active volcanoes, it immediately comes to mind if there is volcanic outgassing. This seems quite obvious as possible source and the total lack of discussion of this is surprising. Related to this

the back trajectory discussion and figure only going back a few hours is a little surprising? What is the rationale to not look back further and more at regional transport. Both these things might be related and could/should be better discussed.

That is a good point. Indeed, volcanic outgassing is an important source of SO2 that can have an impact on the levels of sulphate observed. Several studies have provided evidence that high SO2 concentrations and high radiation levels facilitate the formation of large amounts of  $H_2SO_4$ , which in turn contribute to particle formation (Foucart et al., 2018; Hyvönen et al., 2005; Mikkonen et al., 2006; Petäjä et al., 2009). The Maïdo observatory can be on the pathway of sporadic SO2 volcanic plumes emitted from the "Piton de la Fournaise" volcano, located to the south of the island (Foucart et al., 2018).

The presence of volcanic eruptions is commonly observed at Maïdo, with SO2 concentrations reaching up to several hundred of ppb (Figure A1, Foucart et al., 2018). In the work of Foucart et al. (2018), the presence of the volcanic emissions (plume) was considered to be present when the SO2 concentration at Maïdo reached values higher than 1 ppb (hourly average). Even though volcanic eruptions were not observed during our field campaign, we have analysed the SO2 concentrations observed at Maïdo during this period. As it can be noted in the figure below, SO2 levels rarely overpassed 0.4 ppb concentrations, and only two maximum episodes higher than 1 ppb were observed on March 16 and March 20-21, outside the cloud samples collected (blue stars). However, also if SO2 levels were quite low during the field campaign, their contribution to the sulphate formation cannot be ignored.

We have included this discussion in the manuscript as follows:

"SO2 emissions associated with volcanic eruptions have been already evaluated at La Reunion. Foucart et al. (2018) have shown that SO2 levels can reach up to 500 ppb at Maïdo Observatory during volcanic activity. No volcanic eruptions were reported during our field campaign, and the observed SO2 average concentrations were 0.19  $\pm$  0.24 ppb. Despite those low values, some higher concentrations (up to 1.2 ppb) were reported in days outside cloud sampling. Thus, the contribution of SO2 in the formation of sulphate can be neither affirmed nor ignored".

Regarding back trajectories, we have decided to make an agreement between the results we wanted to explore and the computing time to do that. Our first insights into low-spatial resolution back trajectories have shown that air masses are transported over the ocean and at high altitudes for almost all the cloud events (except for R10A event). As it can be noted in Figure 10, all the back trajectories are over the ocean 12 hours (or less) before the arrival at the sampling point.

In this study, we have decided to design the analysis based on the land-atmosphere interactions and potential effects on the cloud composition from local sources from the island. Due to the location of the island, far away from the continents, our analysis focused on high resolution back trajectories over the island over a short period rather than low resolution back trajectories over a long period.

We have included this discussion in the manuscript as follows:

"Our first insights into low-resolution back trajectories have shown that air masses are transported over the ocean and at high altitudes for almost all the cloud events (except for R10A event). Almost all the back trajectories were over the ocean 12 hours (or less) before the arrival at the sampling point (Piste Omega). We have then prioritized a high-resolution approach over the island and over a short period to evaluate land-atmosphere interactions and potential effects on the cloud composition from local sources. Therefore, to estimate the influence of the soil type located under the trajectory points, an interpolation of the land zones was done using the Corinne Land Cover 2018 inventory (UE – SOeS, CORINE Land Cover, 2018, Geoportail, https://www.geoportail.gouv.fr/)."

The experimental section lacks critical details, if this is a clear description of the collector and how the size cut was determined or a clear discussion on blank values (both organics and metals as the authors used an aluminum collector). The blank discussion is a must! For all compounds even the gas phase ones.

We agree with the reviewer. A description of the cloud collector used in this field campaign is added to the SI file.

Cloud collector description:

"The cloud sampler used in this study is a newly designed collector for sampling cloud droplets suitable for cloud chemical and microbiological analysis. It is based on the same impaction procedure as the CWS sampler (Kruisz et al., 1993) but using 3 vertical impaction plates instead of one. CWS samplers are commonly used by our team for collecting cloud water at the PUY station. The inlet width and the distance between the inlet and impaction plate are conserved to keep the same cut-off diameter around 7 µm as estimated by Kruisz et al. (1993). This impactor is made in aluminium that is easily sterilisable for biological analysis. It is composed of 3 parts (inlet, impaction plates, collection funnel) that are installed on a metallic box where a ventilator fan is installed, before the sampling. These parts are sterilized before each sampling. The sampler runs with a 12V battery, and the total mass of the system is around 8 kg, allowing to install it at the top of the 10 m mast. Those developments are detailed in a paper that will be submitted soon (Vaïtilingom et al., in prep, 2021)."

**Cloud water analysis:**

The cloud water collector was firstly cleaned with MilliQ water and let dry under a fume hood. Once dried, the three pieces were placed in sterilization bags with clean nitrile gloves and then sterilized to avoid microbiological contamination. The pieces were extracted from sterilization bags and mounted with sterile gloves immediately before sampling. Chemical and microbiological contamination were tested at the beginning of the campaign, by spreading sterilized MilliQ water on the clean cloud collector and analyzing it with different approaches. This blank sample will therefore be named "blank" in the following text.

Since the cloud collector is made of Aluminium, we brought particular attention to metal contamination. The blank correction was taken into account for trace metal analysis, where the concentrations found in the blank were subtracted to the concentrations measured in the samples. Similarly, Fe(II) and Fe(III) concentrations were measured in the blank and subtracted for the samples. The metal concentrations found in the blank were similar to the ones already measured for previous campaigns at the puy de Dôme station.

Analogously, the concentration of main inorganic and organic ions, measured by ion chromatography, resulted to be similar to the blank concentration measured previously at the puy de Dôme station and negligible compared to ions concentration in the samples. Microbiological contamination was tested by a culture of 100  $\mu$ L of blank on R2A and no microbial colonies were observed after 7 days at 15°C.

**Gas-phase measurements**

The procedure of tube conditioning for low-soluble gaseous hydrocarbons has been described in Wang et al., 2019. During the BIOMAIDO field campaign, we paid attention to the quality of the blanks by analyzing on a regular basis non-exposed sealed tubes, but that were transported and kept sealed during the field campaign. Our analysis showed that all compounds including monoaromatic compounds excepted toluene were below 5 ppt; the one of toluene was 20 ppt which is lower than the observed ambient concentrations (table S6).

For polar gaseous OVOC, as described in the Supplement Information, the PFBHA tubes were tested at the laboratory during the optimization step of the sorbent coating and derivatization phase with PFBHA. After the coating step, we checked whether there was no contamination in derivatized OVOC oximes. During the field campaign, some pre-coated tubes (2) with PFBHA were not exposed and stored at 4°C. Those tubes were analyzed at the laboratory following the same procedure as the ambient tubes; the corresponding OVOC concentrations were subtracted to the ones from the ambient ones when needed.

**A sentence was added in the text:**

"Chemical and microbiological contaminations were tested at the beginning of the campaign, by spreading sterilized MilliQ water on the clean cloud collector and analyzing it with different approaches. Contaminant's concentration was subtracted to sample concentration in the case of trace metal analysis, while it was considered to be negligible for main inorganic and organic ions and microbial concentration."

The authors tackle a very challenging task of measuring H2O2 and iron speciation in cloud samples. Given that both species are highly reactive, it is critical to specify how long the samples were sitting before being aliquoted and worked up. See e.g. Siefert et al., 1998 for measurements at 10 minutes. Given the reactivity, even long collection times will lead to reactivity in the sampling bottle. This is a little acknowledged in the discussion but it is unclear how long this was. Depending on the time delay that whole section could be not informative when the samples were sitting too long to say anything on the concentrations in clouds and the current text says this a little with the disclaimer in it. Therefore may be that section can be

omitted as the experiment might not allow for any clear statement (and you have a lot of other interesting high quality observations).

The manuscript has been updated following your concerns. Iron speciation measurements are presented in the manuscript and  $H_2O_2$  quantification is now in the SI section.

Concerning iron speciation measurements, the samples have been conserved under frozen conditions until the analysis. Before the analysis, ferrozine is added to the aliquot. These measurements have been performed by our teams in the past 20 years.

Concerning  $H_2O_2$ , as pointed in the original version of the manuscript, we have some concerns about the measurements since the aqueous samples have been stored under frozen conditions before arriving in the lab where the analysis must be performed. These frozen samples were melted, and the solutions were analysed by UV-Visible spectroscopy using phydroxyphenylacetic acid (HPAA, purity > 98 %) and horseradish peroxidase (POD). We know that part of the  $H_2O_2$  can be lost and the  $H_2O_2$  quantification is surely underestimated. The analysis of  $H_2O_2$  should be thus considered questionable and is now indicated in the SI.

Statistics are being used but they need to be described in the experimental section. For the correlations it is critical to say what is statistically significant and what not and at what confidence level. This is never specified and often only r2 values are given which have no direct meaning while the discussion is qualitative "strong correlation".

We agree with the reviewer about the validity of the statistics. With only 14 samples, p-values are important. Except for the correlation between acetate and formate (with a p-value = 0.057), the other p-values are below 0.05. To highlight this precision, we have included the p-values for each correlation discussed in the manuscript.

| p-values (Pearson):                      |         |       |         |         |         |         |         |       |         |         |
|------------------------------------------|---------|-------|---------|---------|---------|---------|---------|-------|---------|---------|
|                                          |         |       |         |         |         |         |         |       |         |         |
| Variables                                | Na+     | NH4+  | K+      | Mg2+    | Ca2+    | Cl-     | NO3-    | SO42- | Cations | Anions  |
| Na+                                      | 0       | 0.011 | 0.004   | <0,0001 | 0.001   | <0,0001 | <0,0001 | 0.048 | <0,0001 | <0,0001 |
| NH4+                                     | 0.011   | 0     | 0.047   | 0.026   | 0.016   | 0.002   | 0.000   | 0.030 | 0.005   | 0.001   |
| K+                                       | 0.004   | 0.047 | 0       | 0.053   | <0,0001 | 0.001   | <0,0001 | 0.062 | 0.001   | 0.000   |
| Mg2+                                     | <0,0001 | 0.026 | 0.053   | 0       | 0.011   | 0.000   | 0.003   | 0.027 | <0,0001 | 0.000   |
| Ca2+                                     | 0.001   | 0.016 | <0,0001 | 0.011   | 0       | 0.000   | <0,0001 | 0.012 | <0,0001 | <0,0001 |
| CI-                                      | <0,0001 | 0.002 | 0.001   | 0.000   | 0.000   | 0       | <0,0001 | 0.049 | <0,0001 | <0,0001 |
| NO3-                                     | <0,0001 | 0.000 | <0,0001 | 0.003   | <0,0001 | <0,0001 | 0       | 0.019 | <0,0001 | <0,0001 |
| SO42-                                    | 0.048   | 0.030 | 0.062   | 0.027   | 0.012   | 0.049   | 0.019   | 0     | 0.026   | 0.018   |
| Cations                                  | <0,0001 | 0.005 | 0.001   | <0,0001 | <0,0001 | <0,0001 | <0,0001 | 0.026 | 0       | <0,0001 |
| Anions                                   | <0,0001 | 0.001 | 0.000   | 0.000   | <0,0001 | <0,0001 | <0,0001 | 0.018 | <0,0001 | 0       |
|                                          |         |       |         |         |         |         |         |       |         |         |
|                                          |         |       |         |         |         |         |         |       |         |         |
| Coefficients of determination (Pearson): |         |       |         |         |         |         |         |       |         |         |
|                                          |         |       |         |         |         |         |         |       |         |         |
| Variables                                | Na+     | NH4+  | K+      | Mg2+    | Ca2+    | Cl-     | NO3-    | SO42- | Cations | Anions  |
| Na+                                      | 1       | 0.430 | 0.509   | 0.904   | 0.641   | 0.874   | 0.755   | 0.287 | 0.984   | 0.863   |
| NH4+                                     | 0.430   | 1     | 0.290   | 0.348   | 0.397   | 0.549   | 0.650   | 0.337 | 0.500   | 0.615   |
| K+                                       | 0.509   | 0.290 | 1       | 0.276   | 0.896   | 0.643   | 0.742   | 0.261 | 0.608   | 0.699   |
| Mg2+                                     | 0.904   | 0.348 | 0.276   | 1       | 0.427   | 0.668   | 0.525   | 0.345 | 0.857   | 0.658   |
| Ca2+                                     | 0.641   | 0.397 | 0.896   | 0.427   | 1       | 0.717   | 0.887   | 0.419 | 0.743   | 0.812   |
| CI-                                      | 0.874   | 0.549 | 0.643   | 0.668   | 0.717   | 1       | 0.847   | 0.285 | 0.906   | 0.975   |
| NO3-                                     | 0.755   | 0.650 | 0.742   | 0.525   | 0.887   | 0.847   | 1       | 0.378 | 0.840   | 0.932   |
| SO42-                                    | 0.287   | 0.337 | 0.261   | 0.345   | 0.419   | 0.285   | 0.378   | 1     | 0.349   | 0.387   |
| Cations                                  | 0.984   | 0.500 | 0.608   | 0.857   | 0.743   | 0.906   | 0.840   | 0.349 | 1       | 0.920   |
| Anions                                   | 0.863   | 0.615 | 0.699   | 0.658   | 0.812   | 0.975   | 0.932   | 0.387 | 0.920   | 1       |

The PLS method needs to be clearly described. The whole PLS discussion is not very clear, neither how PLS was performed (experimental?), nor the results. This is very obscure actually. Also unclear is if PLS does fine with non-normal distribution of variables and when the variables are not independent such as LWC and chemical concentrations. I do not say anything is wrong, it just need explanation and may be evaluation of this really adds anything to the manuscript?

The partial least square (PLS) regressions are performed using Excel XLSTAT software (Addinsoft, 2021). PLS regression allows summarizing a set of correlations, by grouping the coefficients of determinations (r) in a matrix: the correlation matrix. The parameters studied are, on the one hand, the explanatory variables (e.g., "land use cover"), and on the other, the dependent variables (e.g., chemical concentrations in cloud water).

The studied correlations can be highlighted, either by colouring the correlation matrix (Table S10: Red: correlations; Blue: anti-correlations) or with the correlation map (Figure S9).

We are aware that this graph is more difficult to interpret because it contains an additional dimension that allows observing any correlations between the samples and the parameters (X: explanatory or Y: dependents). PLS has advantages over other techniques when analyzing small sample sizes or data with non-normal distribution (Chin and Newsted, 1999, page 337).

To clarify the use of this statistical tool, we modified the text in the manuscript as follows:

"To this end, partial least square (PLS) regressions are performed using Excel XLSTAT software (Addinsoft, 2021). The PLS approach is a statistical method for modelling complex relationships between explanatory variables (the "Xs") and dependent variables (the "Ys"). Furthermore, PLS regression is adapted for particular data conditions such as small sample sizes or data with non-normal distribution (Chin and Newsted, 1999, page 337).

Hereafter, PLS allows establishing the correlations between the chemical categories, microphysical parameters, and the land use cover. The matrix of the Xs is composed of the LWC matrix and the "land use cover" matrix provided by interpolation on the back-trajectory points (Section 2.4, Figure 10). Percentages in Table S9 have been calculated considering back trajectory points lower than 500 m above sea or ground level and then correlated with the four mainland cover categories to obtain the relative contribution of each area. The matrix of the Ys gathers four groups of compounds (individual concentrations of inorganic ions, (di)carboxylic acids, amino acids, and sugars)."

The quality of the modelling of the PLS regression is evaluated by an index, the  $Q^2$ , which must be positive for the PLS regression to be predictive. Here, the PLS regression is therefore not predictive ( $Q^2 = -0.078$ ). Unsurprisingly, the complexity of the cloud is too great to be determined solely by the "land use cover" at the island level.

Nevertheless, the PLS regression helps to identify the positive (or negative) influences of certain areas ("Farming area", "Vegetation") on chemical concentrations. These correlations are highlighted by the correlation map and the correlation matrix.

To clarify the discussion, we modified the text as follows:

"The index of the predictive quality of the model is slightly negative ( $Q^2 = -0.078$  with one component) which means the model is not predictive. Indeed, cloud chemical composition can be modulated by many other parameters than the chosen explanatory variables, related to the air mass history calculated by the CAT model. Additionally, the cloud chemical composition depends on local microphysics (Möller et al., 1996; Moore et al., 2004; Wieprecht et al., 2005), as well as proximity to sources (Collett et al., 1990; Gioda et al., 2013; Kim et al., 2006; Watanabe et al., 2001), biological activity (Bianco et al., 2019; Vaïtilingom et al., 2013; Wei et al., 2017), seasons (Bourcier et al., 2012; Fu et al., 2012; Guo et al., 2012; Shapiro et al., 2007), and diurnal cycles (Kundu et al., 2010).

In addition, the collinearity (Figure S9) between the average concentration of ions implies that the "Land use cover" does not preferentially influence one ion over another. Categorization of clouds as performed in Renard et al. (2020) is therefore not possible. For instance, marine category (with predominant Na+ or Cl-) or continental one (with predominant NH4+ or NO3-), cannot be proposed for the collected samples since at Reunion Island a "well missed" distribution of ions is observed. This suggests that either air masses had the same history, or, more likely, the presence of physical phenomena leads to the "homogenization" of air masses. Furthermore, on the scale of a small island the use of a very fragmented inventory, does not show any substantial improvement in the understanding of the variability of the chemical content of the collected cloud events.

Nevertheless, one main trend emerges from the correlation matrix (PLS Table S10). "Farming area" is correlated with chemistry, in particular with amino acids (Rmean = 0.39), sugars (Rmean = 0.40), and dicarboxylic acids (Rmean = 0.39). Note that these correlations are slightly overestimated due to the weak anti-correlation (RLWC = -0.27) between the "Farming area" and the LWC. This tendency requires further investigations."

The discussion of the partitioning makes one wonder about analytics. Terpenes are really hard to measure by grab and analyze methods. Could the discussion be clearer on how well the analytics did perform and if there could be realistic error bars on these measurements. Again there is also a concern for storage of the samples and transport? (if not analyzed at Reunion) and potential artifacts.

As explained in the SI, the method used follows the one developed by Wang et al. (2020). Stir Bar Sorptive Extraction (SBSE) was used to pre-concentrate dissolved VOCs in cloud water.

VOCs have been already detected and analysed by this method for different environmental media like river water, seawater, soil, food and flavour (Alves et al., 2005; Coelho et al., 2009; Tredoux et al., 2008), and recently in cloud waters at low concentrations (Wang et al., 2020).

SBSE is an application of stir bars coated with polydimethylsiloxane (PDMS), also called "Twisters". This equilibrium technique leads to the extraction of solutes from the aqueous phase to the PDMS phase, which is controlled by the partitioning coefficient of the solutes between the PDMS phase and the aqueous phase. The partitioning coefficient is usually approximated to the octanol-water partitioning coefficient Kow. The extraction efficiency (E), which corresponds to the recovery of analytes from the samples, depends not only on the Kow but also on the volume of water and the volume of PDMS. At a given KOW, a theoretical E can be determined from different volumes. In practice, other parameters are known to affect the solid-water equilibrium and the ex- traction efficiency as well (Kawaguchi et al., 2005; Ochiai et al., 2001; Pang et al., 2011; Portugal et al., 2008): extraction time, PDMS volume, sample volume, and ionic strength. These factors have been optimized on modelled aqueous matrices to achieve optimal extraction of VOCs. Moreover, the thermo-desorption step by the TD/GC-MS was optimized to guarantee the complete transfer of the studied compounds from the twister to the GC column. Thermo-desorption conditions are the same as for cartridges, which allows total desorption for extracted compounds by Twisters. The SBSE samples were then transferred into an empty cartridge following the same analytical procedure that the one described for gaseous samples.

All the details about performances, methodology, optimisation, and uncertainties associated with the analytical techniques are fully described in the work of Wang et al. (2020). We have included some details and uncertainties in the SI ("VOCs in cloud water") and Table S6.

**Storage details are described in a comment below.**

While I recognize that this manuscript is not a review paper, the results could however be put better in context. There is a substantial literature that is being missed on many of the chemical parameters discussed, on partitioning (going back to the 1980s) and on marine cloud observations. Too often Puy de Dome or source apportionment in metropolitan France seems to be the primary reference in discussions and while the authors might be most familiar with this, it is not necessarily the most appropriate references for context or insights. Overall the referencing could also be improved to justify methods (e.g. cloud collector or HPLC-PAD method) as the authors cite their papers but not where the collector is described in detail or the method but just papers where they use the device/method. The actual primary source would be most useful.

Thanks for this comment. We are aware that other papers can be cited in the manuscript. We would like first to mention that older papers on the cloud chemical composition has been added in the introduction of the manuscript. Concerning the PUY station, this station is mainly under the marine influence that can justify the fact that it is often cited in the text. Moreover, the same kind of measurements are performed on both sites (Reunion Island and PUY). New discussions and literature were added to the manuscript to improve the scientific context of our study. They are listed below:

**Marine sites:**

"The contribution of light acids has also been observed in recent studies in marine environments, where those species dominate the organic contribution to TOC (Boris et al., 2018; Stahl et al., 2021)."

**Air/water partitioning studies:**

"Previous studies had already reported some deviations for the expected phase partitioning equilibrium for small carboxylic acids (Facchini et al., 1992; Winiwarter et al., 1994)."

As described above, the cloud collector has been designed based on a well-known cloud collector (CWS) allowing to collect efficiently cloud water, light enough to be installed on a 10 m mast, easy to be sterilized that is crucial for chemical and biological analysis and running using a 12V battery. This is now described in the SI, section 1.

The HPLC-PAD method has been developed over the last years for the detection and quantification of sugars in aerosol samples. To the best of our knowledge, this is the first time that such wide speciation of sugar analyses has been performed in cloud samples. Therefore, the HPLC-PAD method has been used for the determination of sugars in our study. Full details about it can be found in the work of Samake et al. (2019) and are also described in the SI.

Minor

**Experimental:**

Given the presence of sulfur and aldehydes, would you know if HMSA shows as formaldehyde or not with your analytical method? There is a question if you have reactive sulfur (SO2) that some of if could be in adducts and then if you still determine formaldehyde as formaldehyde or not, idem for sulfate. This will impact both your carbonyls and your sulfate values.

The reviewer is right. In cloud droplets, carbonyl compounds (in particular formaldehyde) form adducts with dissolved SO2. For instance, significant amounts of hydroxymethanesulfonate (HMSA), the adduct of HCHO, have been measured in the cloud or fog water where it constitutes a reservoir for both S(IV) species and HCHO (Ang et al., 1987; Munger et al., 1984). HMSA and more generally S(IV)-carbonyl adducts form rapidly and are stable towards dissociation in acidic media such as cloud droplets (Dasgupta et al., 1980; Munger et al., 1984).

In this work, the derivatization of carbonyls by DNSAOA was performed in an acidified solution of the cloud sample (pH 2.8, related to aniline chlorhydrate 0.1 M used here). Dissociation of the carbonyl-S(IV) adducts in the course of the derivatization reaction are therefore unlikely, and carbonyl concentrations reported in this paper represent the free carbonyl concentrations ([carb]free) and therefore lower estimates of the total carbonyl concentrations, with [carb]tot =[carb]free+[S(IV)-carb] (Deguillaume et al., 2014).

We have included this information in the description of the technique in the supplementary information (SI 1 – carbonyls - LC system and dual fluorescence / mass spectrometry analysis).

A clear description of sample conservation (fridge, freezer, bactericide) is missing.

We agree with the reviewer that some details about conservation are missing in the manuscript.

In the material and methods section, we indicate that all the cloud samples were immediately filtered after collection to remove microorganisms.

Additional information on sample conservation after the sampling and before the analysis are given in the SI as following:

**"Sample conservation:**

During the campaign period, the sample were filtered and stored under frozen or refrigerated conditions based on storage procedures (as the one performed for cloud waters sampled at the PUY station, regularly).

Once the field campaign had finished on 4 April 2019, cloud samples were immediately expedited to France (mainland, Clermont-Ferrand, and Grenoble laboratories) following the storage conditions for each type of analysis, under frozen or refrigerated conditions. This was performed by the transportation and logistics unit of the CNRS (Ulisse). Targeted chemical analyses were performed approximately 10 days after the arrival of the samples at the laboratories."

The authors mention in the ferrozine method description that they used aspartic acid. This seems highly unusual as iron is typically reduced by ascorbic acid? Is aspartic acid a common reducing agent?

The detailed method is correctly described in the supplementary information; however, we have made a mistake in the manuscript. The correct reducing agent is ascorbic acid, and the text was corrected accordingly.

**Observations:**

One misses a discussion of pH? And context to recent studies (e.g. Pye et al., ACP 2020)

We agree with the reviewer. pH is a central component of aqueous chemistry (Pye et al., 2020). pH measurements were performed for each cloud sample. The pH values obtained during this study ranged from 4.7 to 5.5, with average values of 5.25. As it can be noted, the pH values present a weak variability within cloud events.

We added the following paragraph in the manuscript:

"pH is a central component in cloud aqueous phase chemistry since it controls mass transfer and aqueous reactivity. pH measurements were performed for each cloud sample. The pH values obtained ranged from 4.7 to 5.5, with average values of 5.25. As it can be noted, the pH values present a weak variability within cloud events. Uptake of gaseous carbon dioxide is a crucial factor governing cloud pH, especially for an area far from anthropogenic sources (remote environment). Considering the mean atmospheric CO2 level at 298K, the calculated droplet pH is around 5.6. Clouds samples at Reunion Island are therefore a little bit less acidic than predicted by CO2 level. Cloud pH is mainly controlled by sulfuric and nitric acids that are strong inorganic acids and by ammonia that is the most abundant base (Pye et al., 2020). Those compounds can be the results of several processes in cloud water (scavenging of particles, aqueous phase production, uptake from the gas phase). Even if NH4+ leads to a basification of the cloud water, the presence of inorganic and organic acids tends t to be weakly acidified of the water. Other ions such as Cl-, Na+, Mg2+, K+ and Ca2+ modulate H+ and the presence of weak acids (such as formic acid) can also lead to acidity buffering (Tilgner et al., 2021); this highlights the complexity of multiphasic physico-chemical processes controlling cloud water acidity."

Many other studies exist in marine environments. See airborne cloud observations by Sorooshian and others which have organic acids, carbonyls, discussion of chemistry (see current ACPD paper by Stahl et al., 2021 and references therein https://acp.copernicus.org/preprints/acp-2021-403/) or a recent study by Boris et al., 2018 which has many of the species that are covered here at a coastal site or Hatchings et al., 2009

has cloud VOC data, just some examples of actual relevant observations). These are just some examples, there are many more missed observational studies.

Yes, the reviewer is right. The references you mentioned were added in the discussion of the results. Those modifications are briefly recalled below:

Discussion on carboxylic acids:

"The contribution of light acids has also been observed in recent studies in marine environments, where those species dominate the organic contribution to TOC (Boris et al., 2018; Stahl et al., 2021).

Also, a recent airborne study by Stahl et al. (2021) in Southern Asia has shown TOC concentrations ranging from 0.018–13.66 mg C L-1.

Carboxylic acids were also dominant contributors in the TOC concentrations obtained during CAMP2Ex campaign in southeast Asia, where acetate, formate, and oxalate accounted for 23% of the TOC measured (Stahl et al., 2021)."

Discussion on low solubility VOC:

"Similarly, aromatic compounds (toluene, ethylbenzene, and xylenes) were detected in clouds obtained in northern Arizona. Average concentrations were 1.02, 1.08, 1.20, and 0.54 µg L-1 for toluene, ethylbenzene, m+p-xylene, and o.xylene, respectively, contributing less than 1 % to the dissolved organic carbon (Hutchings et al., 2009)."

Discussion on sugars:

"The concentrations of total sugars in our cloud events ranged from 10.30 to 66.50 µmol L-1 (average of 22.18  $\pm$  15.40 µmol L-1 (121.3  $\pm$  69.56 ng m-3)). These total concentrations are higher than those obtained for aerosol studies performed in Chichijima Island (46.7 ± 49.5 ng m-3; Verma et al., 2018) and in Okinawa Island (62.0  $\pm$  54.9 ng m-3; Zhu et al., 2015) both located in the western North Pacific. The higher ambient concentrations could be related to the presence of more biogenic sources at Reunion Island. Regarding the contribution of sugar species to the total average, we found that polyols (sugar alcohols) present the higher contribution (52 %) followed by glycerol (31 %) and glucose (13.7 %). Sorbitol presents the highest fraction, followed by glycerol, glucose, and mannitol. The study of Boris et al. (2018) has also analysed some sugars species in fog samples measured in the Southern California Coast. However, fewer sugar species have been detected, with being levoglucosan the highest contributor (0.04 µmol L-1; Boris et al., 2018). The profile observed in cloud water at Reunion Island is quite dissimilar to that observed in aerosol measurements, where glucose, mannitol, and arabitol present the most abundant concentrations in France (Samaké et al., 2019), in Chichijima Island (Verma et al., 2018) and Okinawa Island (Zhu et al., 2015). Thus, this result suggests the presence of additional sources rather than aerosols in the cloud water, but their characterisation needs to be further investigated."

As for partitioning discussion, there were substantial discussion on small molecular weight organics and their partitioning in fog all the way back to the 1980s, see e.g. Winiwarter et al., 1994 and many papers... to the present day. see Stieger et al.,2021. Overall this discussion is quite superficial in the present manuscript. Other authors looked even at droplet size resolved differences.

We agree that several other studies have studied the partitioning of organics in clouds/fogs. We added several references in the discussion of the results to follow your comment:

"Previous studies had already reported some deviations for the expected phase partitioning equilibrium for small carboxylic acids (Facchini et al., 1992; Winiwarter et al., 1994)."

The study from Steiger et al. (2021) is interesting but is more linked with the partitioning of gases between the gas phase and the aerosol phase (dry or deliquescent particles).

Details

L35 "As expected, our findings show the presence of compounds of marine origin in cloud water samples (e.g., chloride, sodium) demonstrating ocean–cloud exchange" this is a non statement, as any cloud water will have Cl and Na,. maybe say something on the ratio but not a sentence that does not say anything. Overall the abstract is lacking quantitative information.

We followed the reviewer's comments regarding the marine origin of the cloud samples and added more quantitative information in the abstract.

L45 "Additionally, several VOCs (oxygenated and low-soluble VOCs) were analysed in both gas and aqueous phases." But what was the outcome, quantitative information is missing here.

New quantitative information was added for VOCs quantification in the abstract.

L64 how do clouds impact homogeneous gas phase chemistry? Consider reformulating

The text has been reformulated as suggested.

L70 What is the rationale here for the late 1990s.? this is a little unfair to some of the early studies who looked at organic matter in clouds... carbonyls, organic acids and even VOCs and higher organics were studies way before by people like Capel, Munger, Collett, Fuzzi and others (see also early EU funded large studies at Great Dunn Fell, Kleiner Feldberg or Po Valley.... Including papers on Henry's law).

We understand your comment regarding older studies on organic matter in clouds. In the introduction, our research context merely shows the increase of work in the cloud chemistry area in the last 15 years. We recognize that references selected in the introduction do not fully cover all the literature in the field. However, we think that the references suggested represent a good selection to put our work in context. We also estimate that more discussion and references will make the introduction more extensive but less fluent to read.

To follow your comment, we have added two new sampling sites/experiements that we have forgotten to mention in the introduction: the Great Dun Fell in England (Choularton et al., 1997) and the Kleiner Feldberg in Germany (Fuzzi et al., 1994; Wobrock et al., 1994).

The statement on the non-targeted compounds, there are some (semi-)quantitative papers out there using chromatographic separations by Decesari or Herckes while for mass spectrometry and besides your work, there are others too who used this like Mazzoleni (https://doi.org/10.1021/es903409k).

Yes, this is true. The reference has been added to the manuscript.

**Also geographically Southern Hemisphere, there is work in Namibia and other locations.**

We looked carefully at the literature if field experiments have been performed at Namibia looking at cloud or fog chemistry. We found that the AEROCLO-sA experiments have been performed in 2017 and that fog water chemical composition has been investigated (see the abstract online: https://meetingorganizer.copernicus.org/IFDA2019/IFDA2019-142.pdf). But we did not find a publication that presents this work. A work has been also performed in Namibia by Eckardt and Schemenauer, 1998. This study gives information on the ion concentrations and ion enrichment relative to sea water, in Namib Desert fog water. They showed high concentration of sulphate due to marine sources. This article is now cited in the manuscript in section 3.2.

Regarding other sites in the Southern Hemisphere, the study from Verhoeven et al., (1987) looked at the fog chemistry in New Zealand. Concentrations are rather low and ions come mainly from sea salt. The pH is also mainly controlled by dissolved CO2. This reference is now cited in section 3.2. Finally, we found the study from Beiderwieden et al. (2005) that followed fog chemistry at the eastern Andes cordillera. This article is not cited in the manuscript.

The following details have been corrected in the manuscript:

L86" "near urban conglomerates"? does not sound right? Consider reformulating

L185 Typo in sulfate SO42- not -2

L268 and others: Deff may be write D eff with eff as subscript

L288 and other locations correlations please state what is significant and what not and at what level.

p-values were added for all the correlations described in the manuscript

L313 +- 44.0 please keep decimals consistent

L340 "contrarily" sounds odd starting the sentence with an adverb, consider reformulating

L356 even though.... Consider reformulating

357" what does "is found to be dominant" mean?

L370 If you keep this discussion then the issue of storage and possible artifact form reactivity before you measure needs to be front and center and not just some detail at the end

L478 AA\_ contribution

L480 cloudS

L488 AA\_ distribution

L512 are all your OVOCs carbonyls?

Yes, all the OVOCs are carbonyls.

**L517 and other locations. What are "highly marine" clouds at PUY?**

The "highly marine" clouds were defined in the study of Renard et al. (2020), by using agglomerative hierarchical clustering (AHC).

From Renard et al. (2020): "Then, we performed agglomerative hierarchical clustering (AHC), an iterative classification method, the aim of which was to make up homogeneous groups of objects (categories) on the basis of their description by a set of variables (chemical variables, herein) describing the dissimilarity between the objects (cloud events, herein). The AHC produced a dendrogram which showed the progressive grouping of the data. To calculate the dissimilarity between samples, we applied the common Ward's agglomeration method (which minimized the within-group inertia) using Euclidean distance. The data were centered-reduced, to avoid variables with strong variance which unduly weighed on the results. The truncation level was automatically defined on the base of the entropy, and therefore the number of categories to retain."

L522 OH radical: please use center dot symbol

L539 I suggest you say low solubility and VOCs (no "' " when plural)

L541 suggest you say "even THOUGH these compounds..."

L541 what does"Sanitary" mean? Do you mean that they have a potential health effect? Adverse effect on environmental or human health?

The text has been changed accordingly.

L552 suggest you cut one decimal in the numbers idem lines 562/563

L563 " in his review" suggest to use "their" review this was more than one author

L675 what is a "cadastre"?

The word has been changed by "inventory".

L688: "which could suggest the influence of dust sources (Samaké et al., 2019b)." but how does Ca look in these samples, given what you say about Ca, does this here really make sense?

As discussed in the inorganic ions section, an enrichment of calcium relative to seawater is observed in our measurements. The excess of Ca2+ was already observed in cloud water (Benedict et al., 2012; Straub et al., 2007), which may be associated with the mineral soil contribution. In addition, the presence of saccharides in soil or mineral dust has also been reported in previous studies, as well as the correlation of sugar species and calcium (Liang et al., 2016; Simoneit et al., 2004; Zhu et al., 2015).

We have added more discussion to clarify this section:

"Strong correlations are observed for glucose and most of the polyol species ( $r^2$ = 0.69 -0.80, p-value: 0.001-<0.0001) with calcium. The correlation between saccharides and calcium has been already observed in previous aerosol studies, associated with the influence of mineral dust from soils (Liang et al., 2016; Samaké et al., 2019; Zhu et al., 2016). The presence of soil

or mineral dust can be related to resuspension processes and therefore contribute to ambient aerosols, adding sugars species such as primary saccharides and polyols (Liang et al., 2016; Simoneit et al., 2004). The correlations between sugars species and calcium obtained in our study could suggest the influence of mineral dust from soils."

L689-91 "Strong correlations are also observed between polyols (inositol, sorbitol, arabitol, and mannitol) with nitrate and potassium, suggesting the contribution from biomass burning sources (Li et al., 2003). Interestingly, levoglucosan, a well-known biomass burning tracer, does not show any correlation with any of these ions." 1) what is a strong correlation? 2) nitrate is not a biomass burning tracer and 3) many studies showed that levoglucosan and K+ are not necessarily well correlated as K can have other sources while levoglucosan can vary by a factor of up to 10 depending on the fuel burnt. The latter is why I mentioned that French soure apportionment studies are not necessarily relevant for what is happening at La Reunion, the biomass burning seen at La Reunion is likely not a fireplace like in the Alps.

We thank the reviewer for this comment. We answer their questions below:

- 1) Correlation coefficients and p-values were added to the discussion.
- 2) Nitrate was found to be a marker of aged biomass burning emissions, that is why we have discussed our results in this sense. But it is also true that the correlation with nitrate could be related to long transport air masses as well.
- 3) We agree with the reviewer that Reunion Island presents different activities and less biomass burning activity than in France. However, during the field campaign, several fuel burnings with cooking purposes have been observed. As potassium and levoglucosan are both known to be burning tracers, we provide the correlations here.

We have added new discussions and references from other studies in the sugar's section and in the environmental conditions one, as follows:

"The concentrations of total sugars in our cloud events ranged from 10.30 to 66.50 µmol L-1 (average of 22.18  $\pm$  15.40 µmol L-1 (121.3  $\pm$  69.56 ng m-3)). These total concentrations are higher than those obtained for aerosol studies performed in Chichijima Island (46.7 ± 49.5 ng m-3; Verma et al., 2018) and in Okinawa Island (62.0  $\pm$  54.9 ng m-3; Zhu et al., 2015) both located in the western North Pacific. The higher ambient concentrations could be related to the presence of more biogenic sources at Reunion Island. Regarding the contribution of sugar species to the total average, we found that polyols (sugar alcohols) present the higher contribution (52 %) followed by glycerol (31 %) and glucose (13.7 %). Sorbitol presents the highest fraction, followed by glycerol, glucose, and mannitol. The study of Boris et al. (2018) has also analysed some sugars species in fog samples measured in the Southern California Coast. However, less sugar species have been detected, being levoglucosan the highest contributor (0.04 µmol L-1; Boris et al., 2018).The profile observed in cloud water at Reunion Island is quite dissimilar to that observed in aerosol measurements, where glucose, mannitol and arabitol present the most abundant concentrations in France (Samaké et al., 2019), in Chichijima Island (Verma et al., 2018) and Okinawa Island (Zhu et al., 2015). Thus, this result suggests the presence of additional sources rather than aerosols in the cloud water, but their characterisation need to be further investigated."

L733"However, our results depict even higher supersaturation of terpenoids, suggesting their importance in the aqueous phase chemistry in highly impacted tropical areas." What is this statement based on?

We agree with the reviewer that this sentence is not very clear. Our results show the cloud concentrations of terpenoids were much higher than predicted by Henry's law, presenting significant deviations from the partitioning theory, which challenges the knowledge about the transfer of species within the gas and aqueous phases. In addition, these deviations seem to be even higher in tropical areas, where the emission of terpenoids is induced by environmental conditions.

We have changed the text as follows:

"However, our results depict even higher supersaturation of terpenoids with concentrations in cloud water much higher than predicted by Henry's law, evidencing a deviation from thermodynamically expected partitioning in the aqueous phase chemistry in this highly impacted tropical area."

On figures:

Figure 4: hard to see the difference sin shades and impossible to read the % numbers in the pie chart and for sure there should be less digits

The figure was updated to make it clearer, and text font has been increased.

Figure5: again hard to see the differences in shading and what is the rationale for the color? As there are anhydrous, there are polyols, there are saccharides... but all are mixes?

The figure was updated to make it clearer and text font has been increased. Sugars are now separated by groups of sugars in the bar chart and by species in the pie chart. A rainbow palette was used to represent different compounds in colours varying between blues and reds.

Figure 6: har dot read the number sin the pie diagram also too many digits. Same thing what is the rationale for the color coding?

The figure was updated to make it clearer, and text font has been increased. A rainbow palette was used to represent different compounds in colours varying between blues and reds.

Figure 7: left panel/ why just lines with dots? Right panel: right pie chart cannot read too small and what are light acids?

Left panel: The lollipop chart is a kind of bar chart to represents totals. We decide to use it instead of a bar chart. The right panel has been updated. Light acids represent the sum of major organic light acids (acetate, formate, oxalate, and lactate). A description has been added to the figure's caption.

**References mentioned**

Alves, R. F., Nascimento, A. M. D. and Nogueira, J. M. F.: Characterization of the aroma profile of Madeira wine by sorptive extraction techniques, Anal. Chim. Acta, 546(1), 11–21, doi:10.1016/j.aca.2005.05.012, 2005.

Ang, C. C., Lipari, F. and Swarin, S. J.: Determination of Hydroxymethanesulfonate in Wet Deposition Samples, Environ. Sci. Technol., 21(1), 102–105, doi:10.1021/es00155a013, 1987.

Benedict, K. B., Lee, T. and Collett, J. L.: Cloud water composition over the southeastern Pacific Ocean during the VOCALS regional experiment, Atmos. Environ., 46, 104–114, doi:10.1016/j.atmosenv.2011.10.029, 2012.

Boris, A. J., Napolitano, D. C., Herckes, P., Clements, A. L. and Collett, J. L.: Fogs and air quality on the Southern California coast, Aerosol Air Qual. Res., 18(1), 224–239, doi:10.4209/aaqr.2016.11.0522, 2018.

Choularton, T. W., Colvile, R. N., Bower, K. N., Gallagher, M. W., Wells, M., Beswick, K. M., Arends, B. G., Möls, J. J., Kos, G. P. A., Fuzzi, S., Lind, J. A., Orsi, G., Facchini, M. C., Laj, P., Gieray, R., Wieser, P., Engelhardt, T., Berner, A., Kruisz, C., Möller, D., Acker, K., Wieprecht, W., Lüttke, J., Levsen, K., Bizjak, M., Hansson, H. C., Cederfelt, S. I., Frank, G., Mentes, B., Martinsson, B., Orsini, D., Svenningsson, B., Swietlicki, E., Wiedensohler, A., Noone, K. J., Pahl, S., Winkler, P., Seyffer, E., Helas, G., Jaeschke, W., Georgii, H. W., Wobrock, W., Preiss, M., Maser, R., Schell, D., Dollard, G., Jones, B., Davies, T., Sedlak, D. L., David, M. M., Wendisch, M., Cape, J. N., Hargreaves, K. J., Sutton, M. A., Storeton-West, R. L., Fowler, D., Hallberg, A., Harrison, R. M. and Peak, J. D.: The Great Dun Fell Cloud Experiment 1993: An overview, Atmos. Environ., 31(16), 2393–2405, doi:10.1016/S1352-2310(96)00316-0, 1997.

Coelho, E., Coimbra, M. A., Nogueira, J. M. F. and Rocha, S. M.: Quantification approach for assessment of sparkling wine volatiles from different soils, ripening stages, and varieties by stir bar sorptive extraction with liquid desorption, Anal. Chim. Acta, 635(2), 214–221, doi:10.1016/j.aca.2009.01.013, 2009.

Dasgupta, P. K., DeCesare, K. and Ullrey, J. C.: Determination of atmospheric sulfur dioxide without tetrachloromercurate(II) and the mechanism of the Schiff reaction, Anal. Chem., 52(12), 1912–1922, doi:10.1021/ac50062a031, 1980.

Deguillaume, L., Charbouillot, T., Joly, M., Vaïtilingom, M., Parazols, M., Marinoni, A., Amato, P., Delort, A. M., Vinatier, V., Flossmann, A., Chaumerliac, N., Pichon, J. M., Houdier, S., Laj, P., Sellegri, K., Colomb, A., Brigante, M. and Mailhot, G.: Classification of clouds sampled at the puy de Dôme (France) based on 10 yr of monitoring of their physicochemical properties, Atmos. Chem. Phys., 14(3), 1485–1506, doi:10.5194/acp-14-1485-2014, 2014.

Eckardt, F. D. and Schemenauer, R. S.: Fog water chemistry in the Namib Desert, Namibia, Atmos. Environ., 32(14–15), 2595–2599, doi:10.1016/S1352-2310(97)00498-6, 1998.

Facchini, M. C., Fuzzi, S., Kessel, M., Wobrock, W., Jaeschke, W., Arends, B. G., Mols, J. J., Berner, A., Solly, I., Kruisz, C., Reischl, G., Pahl, S., Hallberg, A., Ogren, J. A., Fierlinger-Oberlinninger, H., Marzorati, A. and Schell, D.: The chemistry of sulfur and nitrogen species in a fog system a multiphase approach., Tellus B, 44B, 505–521, doi:10.1034/j.1600-0889.1992.t01-4-00005.x, 1992.

Foucart, B., Sellegri, K., Tulet, P., Rose, C., Metzger, J. M. and Picard, D.: High occurrence of new particle formation events at the Maïdo high-altitude observatory (2150 m), Réunion (Indian Ocean), Atmos. Chem. Phys., 18(13), 9243–9261, doi:10.5194/acp-18-9243-2018, 2018.

Fuzzi, S., Facchini, M. C., Schell, D., Wobrock, W., Winkler, P., Arends, B. G., Kessel, M., Möls, J. J., Pahl, S., Schneider, T., Berner, A., Solly, I., Kruisz, C., Kalina, M., Fierlinger, H., Hallberg, A., Vitali, P., Santoli, L. and Tigli, G.: Multiphase chemistry and acidity of clouds at Kleiner Feldberg, J. Atmos. Chem., 19(1–2), 87–106, doi:10.1007/BF00696584, 1994.

Hutchings, J. W., Robinson, M. S., McIlwraith, H., Triplett Kingston, J. and Herckes, P.: The chemistry of intercepted clouds in Northern Arizona during the North American monsoon season, Water. Air. Soil Pollut., 199(1–4), 191–202, doi:10.1007/s11270-008-9871-0, 2009.

Hyvönen, S., Junninen, H., Laakso, L., Dal Maso, M., Grönholm, T., Bonn, B., Keronen, P., Aalto, P., Hiltunen, V., Pohja, T., Launiainen, S., Hari, P., Mannila, H. and Kulmala, M.: A look at aerosol formation using data mining techniques, Atmos. Chem. Phys., 5(12), 3345–3356, doi:10.5194/acp-5-3345-2005, 2005.

Kawaguchi, M., Sakui, N., Okanouchi, N., Ito, R., Saito, K. and Nakazawa, H.: Stir bar sorptive extraction and trace analysis of alkylphenols in water samples by thermal desorption with in tube silylation and gas chromatography-mass spectrometry, J. Chromatogr. A, 1062(1), 23–29, doi:10.1016/j.chroma.2004.11.033, 2005.

Kruisz, C., Berner, A. and Brandner, B.: A cloud water sampler for high wind speeds, in Proceedings of the EUROTRAC Symposium 1992, edited by W. Borrell, P.M., Borrell, P., Cvitas, T., Seiler and 30 (Eds.), pp. 523–525, SPB Academic Publishing., 1993.

Liang, L., Engling, G., Du, Z., Cheng, Y., Duan, F., Liu, X. and He, K.: Seasonal variations and source estimation of saccharides in atmospheric particulate matter in Beijing, China, Chemosphere, 150, 365–377, doi:10.1016/j.chemosphere.2016.02.002, 2016.

Mikkonen, S., Lehtinen, K. E. J., Hamed, A., Joutsensaari, J., Facchini, M. C. and Laaksonen, A.: Using discriminant analysis as a nucleation event classification method, Atmos. Chem. Phys., 6(12), 5549–5557, doi:10.5194/acp-6-5549-2006, 2006.

Munger, J. W., Jacob, D. J. and Hoffmann, M. R.: The occurrence of bisulfite-aldehyde addition products in fog- and cloudwater, J. Atmos. Chem., 1(4), 335–350, doi:10.1007/BF00053799, 1984.

Ochiai, N., Sasamoto, K., Takino, M., Yamashita, S., Daishima, S., Heiden, A. and Hoffman, A.: Determination of trace amounts of off-flavor compounds in drinking water by stir bar sorptive extraction and thermal desorption GC-MS, Analyst, 126(10), 1652–1657, doi:10.1039/b102962m, 2001.

Pang, X., Lewis, A. C. and Hamilton, J. F.: Determination of airborne carbonyls via pentafluorophenylhydrazine derivatisation by GC-MS and its comparison with HPLC method, Talanta, 85(1), 406–414, doi:10.1016/j.talanta.2011.03.072, 2011.

Petäjä, T., Mauldin, R. L., Kosciuch, E., McGrath, J., Nieminen, T., Paasonen, P., Boy, M., Adamov, A., Kotiaho, T. and Kulmala, M.: Sulfuric acid and OH concentrations in a boreal forest site, Atmos. Chem. Phys., 9(19), 7435–7448, doi:10.5194/acp-9-7435-2009, 2009.

Portugal, F. C. M., Pinto, M. L. and Nogueira, J. M. F.: Optimization of Polyurethane Foams for Enhanced Stir Bar Sorptive Extraction of Triazinic Herbicides in Water Matrices, Talanta, 77(2), 765–773, doi:10.1016/j.talanta.2008.07.026, 2008.

[revised manuscript text omitted]